# Geographical variation in hotspots of stunting among under-five children in Ethiopia: A geographically weighted regression and multilevel robust Poisson regression analysis

**Beminate Lemma Seifu**[1]*, **Getayeneh Antehunegn Tesema**[2]*, **Bezawit Melak Fentie**[3], **Tirualem Zeleke Yehuala**[4], **Abdulkerim Hassen Moloro**[5], **Kusse Urmale Mare**[5]

1 Department of Public Health, College of Medicine and Health Sciences, Samara University, Samara, Ethiopia, 2 Department of Epidemiology and Biostatistics, Institute of Public Health, College of Medicine and Health Sciences, and comprehensive specialized Hospital, University of Gondar, Gondar, Ethiopia, 3 Department of Clinical Midwifery, School of Midwifery, College of Medicine & Health Sciences, University of Gondar, Gondar, Ethiopia, 4 Department of health informatics, Institute of Public Health, University of Gondar, Gondar, Ethiopia, 5 Department of Nursing, College of Medicine and Health Sciences, Samara University, Samara, Ethiopia

* beminetlemma1915@gmail.com (BLS); getayenehantehunegn@gmail.com (GAT)

**Data Availability Statement:** The dataset supporting the conclusions of this study is available on a public open-access repository,

## Abstract

### Introduction

Childhood stunting is a global public health concern, associated with both short and long-term consequences, including high child morbidity and mortality, poor development and learning capacity, increased vulnerability for infectious and non-infectious disease. The prevalence of stunting varies significantly throughout Ethiopian regions. Therefore, this study aimed to assess the geographical variation in predictors of stunting among children under the age of five in Ethiopia using 2019 Ethiopian Demographic and Health Survey.

### Method

The current analysis was based on data from the 2019 mini Ethiopian Demographic and Health Survey (EDHS). A total of 5,490 children under the age of five were included in the weighted sample. Descriptive and inferential analysis was done using STATA 17. For the spatial analysis, ArcGIS 10.7 were used. Spatial regression was used to identify the variables associated with stunting hotspots, and adjusted $R^2$ and Corrected Akaike Information Criteria (AICc) were used to compare the models. As the prevalence of stunting was over 10%, a multilevel robust Poisson regression was conducted. In the bivariable analysis, variables having a p-value < 0.2 were considered for the multivariable analysis. In the multivariable multilevel robust Poisson regression analysis, the adjusted prevalence ratio with the 95% confidence interval is presented to show the statistical significance and strength of the association.

accessible online at the Measure DHS website: http://www.measuredhs.com.

**Funding:** The author(s) received no specific funding for this work.

**Competing interests:** The authors have declared that no competing interests exist.

## Result

The prevalence of stunting was 33.58% (95%CI: 32.34%, 34.84%) with a clustered geographic pattern (Moran's I = 0.40, p<0.001). significant hotspot areas of stunting were identified in the west and south Afar, Tigray, Amhara and east SNNPR regions. In the local model, no maternal education, poverty, child age 6–23 months and male headed household were predictors associated with spatial variation of stunting among under five children in Ethiopia. In the multivariable multilevel robust Poisson regression the prevalence of stunting among children whose mother's age is >40 (APR = 0.74, 95%CI: 0.55, 0.99). Children whose mother had secondary (APR = 0.74, 95%CI: 0.60, 0.91) and higher (APR = 0.61, 95%CI: 0.44, 0.84) educational status, household wealth status (APR = 0.87, 95%CI: 0.76, 0.99), child aged 6–23 months (APR = 1.87, 95%CI: 1.53, 2.28) were all significantly associated with stunting.

## Conclusion

In Ethiopia, under-five children suffering from stunting have been found to exhibit a spatially clustered pattern. Maternal education, wealth index, birth interval and child age were determining factors of spatial variation of stunting. As a result, a detailed map of stunting hotspots and determinants among children under the age of five aid program planners and decision-makers in designing targeted public health measures.

## Introduction

Globally, there were 149.2 million stunted, 45.4 million wasted, and 38.9 million overweight children under the age of five in 2020. Except in Africa, the prevalence of stunting among under five children is declining in all regions [1]. In Ethiopia Stunting has steadily decreased, from 58% in 2000 [2] to 36.81% in 2019 [3]. Despite significant improvements, stunting, or poor growth and development, continues to be the most common form of child malnutrition and a major contributor for 28% of child mortality due to malnutrition in Ethiopia [2, 4].

The Sustainable Development Goals (SDGs) target 2.2 aims for ending malnutrition in all its forms [5]. There is an aspiration to achieve, by 2025, a 40% relative reduction in stunting (to a global prevalence of approximately 15%) [5]. Several analysis have shown that despite improvements in child stunting over the last two decades [6], most countries, including Ethiopia, are not on track to reach the SDG 2.2 undernutrition targets [7].

Childhood stunting has both immediate and long-term repercussions, including high child morbidity and mortality, poor development and learning capacity, increased vulnerability for infectious and non-infectious disease [8, 9]. Furthermore, stunting in female children has a negative impact on maternal reproductive outcomes in adulthood [10, 11]. Studies have shown that sex of the child, age of the child, source of drinking water, maternal age and educational status, residence, region, wealth index, birth interval, duration of breast feeding were a significant predictors of stunting among children under the age of five [3, 12–14].

In accordance with studies done on the prevalence and associated factors of childhood stunting, the prevalence of stunting varies significantly throughout Ethiopian regions. The geographical variation of childhood stunting revealed that Amhara, Benishangul Gumuz, Afar, SNNP, and Tigray regions had a significant burden of stunting [15, 16]. Another study

indicated that Stunting was found to be spatially clustered in northern, northwestern, north-eastern, and southern Ethiopia [3].

Previously done studies regarding the spatial distribution of childhood stunting in Ethiopia used the ordinary least square regression [17, 18] and only rural areas of the country were covered [19] and a single study used geographically weighted regression used the 2016 EDHS data [20]. The global statistical (ordinary least square) model assumes the relationship between the dependent and independent variables is identical across the study area. Having said that, the hypothesis of spatial uniformity of the effect of explanatory variables on the outcome variable is often unrealistic [21]. If the parameters vary significantly in space, a global estimator will hide the geographical richness of the phenomenon [22]. Explicitly capturing geographic-based local relationships by examining spatial dependencies and heterogeneity in the relationships between factors and under-five child stunting can aid the government and policymakers in establishing and enacting multiple successful nutrition-specific initiatives. Furthermore, to make informed decisions about preventing and controlling undernutrition, it is important to understand the variability in different areas and the factors that contribute to it. This can help identify gaps in childhood nutrition programs that may not be apparent through regular monitoring of children's nutritional status.

Therefore, this study aimed to explore the spatial relationship between stunting and its determinants among children under the age of five by employing a geographically weighted regression (GWR) analysis using the most recent (2019) EDHS data.

## Method and materials

### Study area, study period and data source

This study was based on the 2019 mini Ethiopian Demographic and Health Survey (EDHS) data. Ethiopia has nine administrative regions: Tigray, Afar, Amhara, Oromia, Somalia, Benishangul-Gumuz, Southern Nation Nationality and People's Region (SNNPR), Gambella, and Harari, as well as two self-governing cities: Addis Ababa and Dire Dawa. EDHS is a nationally representative survey routinely conducted every five years and collects data on basic health indicators like mortality, morbidity, fertility, and maternal and child health-related characteristics. The survey used a two-stage stratified sampling technique to select the study participants. In the first stage, Enumeration Areas (EAs) were randomly selected based on the country's recent population and using the housing census as a sampling frame, households were randomly selected in the second stage. The survey consists of multiple datasets for men, women, children, births, and households. There are different datasets available for each category, providing a comprehensive overview of various aspects related to them. Given that the study population included children under the age of five, we used the Kids Record dataset (KR file) for this study.

The source population of this study was all under-five children in Ethiopia. The study population consisted of all under-five children in the survey. In the current study, a weighted sample of 5,490 children under the age of five were considered for final analysis. Detailed information about the Demographic and Health Survey (DHS) methodology can be found from the official DHS database https://dhsprogram.com/Methodology/index.cfm.

**Study variables.** *Dependent variable.* In this study, the dependent variable was stunting, which was classified as "Yes = 1" for a child with a length or height/age $< -2$ Z score and "No = 0" for a child with a height/age $> -2$ Z score.

*Independent variables.* maternal age, maternal educational status, household wealth index, family size, sex of household head, source of drinking water, sex of the child, age of the child, number of under-five children, place of delivery, type of birth, birth order, preceding birth

interval, duration of breast feeding, antenatal care (ANC) visit, region residence and community maternal literacy level.

*Source of drinking water*. The WHO/UNICEF Joint Monitoring Programmes (JMP) for Water Supply and Sanitation established three drinking water source classifications during the Millennium Development Goals (MDG) monitoring period: improved, unimproved, and other [23].

Improved categories.

- Piped into dwelling

- Piped to yard/plot

- Public tap/standpipe

- Piped to neighbor

- Tube well or borehole

- Protected well

- Protected spring

- Rainwater

- Tanker truck

- Cart with small tank

- Bottled water

  Unimproved categories.

- Unprotected well

- Unprotected spring

- Surface water (river/dam/lake/pond/stream/canal/irrigation channel)

- Other

**Data management and analysis.** Stata version 17 and Arc-GIS version 10.7 were used for data extraction, coding, and analysis. To make the data more representative again, analysis was done using the weighted data. Because the DHS data is hierarchical, the intra-class correlation coefficient (ICC) was calculated to assess the clustering effect. There was a sizable clustering effect, according to the ICC (ICC>10%). A modified Poisson regression with robust standard errors was employed to account for the high prevalence of stunting in the study population. The DHS sampling technique follows a hierarchical structure; therefore, the modified Poisson regression models used (including the bivariate modified Poisson regression) were multilevel mixed effects. In the bi-variable multilevel Poisson regression analysis, variables with a p-value of $< 0.2$ were taken into account for the multivariable analysis.

We have fitted four models separately. Model 1 (null model) was fitted without independent variables to estimate the cluster-level variation of stunting in Ethiopia. Model 2 and Model 3 were adjusted for individual-level variables and community-level variables, respectively. Model 4 was the final model adjusted for individual and community-level variables simultaneously. The deviance (-2Log-Likelihood Ratio (LLR)) was used to test the goodness of model fitness, and the model that had the lowest deviance was the one that best fit the data. After estimating the Adjusted Prevalence Ratio (APR) and its 95% CI, we reported the

significant predictors of stunting among children under the age of five. In the multivariable analysis, only variables with a p-value of 0.05 were regarded to be statistically significant. For the spatial analysis, the outcome and covariates were collected at individual level whereas the geographic coordinate data (latitude and longitude) at cluster level/Enumeration Areas (EAs) level. For the GWR, we have aggregated the outcome and covariates at EA level and then merged with the GPS data using cluster number/EAs as merging variable.

**Spatial analysis.** Global spatial autocorrelation (Global Moran's I) was used to determine if the spatial distribution of stunting among Ethiopian children under the age of five was scattered, clustered, or random [24]. The Global Moran's I statistic is utilized for computing spatial autocorrelation, taking into account attribute values that lie between -1 and +1. When Moran's I value is around -1, it means that stunting among children under the age of five is spread out. On the other hand, a Moran's I value near +1 indicates that stunting is clustered, while a value near 0 means that stunting is randomly distributed across the study area. A statistically significant Moran's I ($p < 0.05$) shows that the distribution of stunting is non-random (either clustered or dispersed) [25]. The Getis-OrdGi* statistics were applied in the hotspot analysis to calculate the GI* statistic for each area to assess how spatial autocorrelation varied across the study geographical area. The Z-score measures and the p-value employed to estimate the statistical significance of clustering. Statistical output with a high GI* indicates a "hot spot", and output with a low GI* indicates a "cold spot" [24].

**Spatial regression analysis.** We conducted Ordinary Least Square (OLS) regression and Geographic Weighted Regression (GWR) to explore the spatial relationship between stunting in under-five children and explanatory variables. The proportion of stunted children among those under the age of five at the Enumeration Area (EA) level used as the dependent variable for spatial regression analysis.

A crucial factor in spatial regression is the neighborhood or bandwidth, which determines the level of smoothing in the model. Choosing the right distance band or number of neighbors for each regression equation can make or break the accuracy of the results. The number of variables in the model, as well as the bandwidth, determine the complexity of a spatial regression model. There are three methods for determining bandwidth: AICc, CV, and bandwidth parameter. In this study, an adaptive kernel was used and the bandwidth was determined by minimizing the AICc value.

*Ordinary least square (OLS) regression*. The ordinary least squares method is a regression model that predicts the association between the dependent and independent variables using a single equation. It assumes that the relationship between the variables is stationary or uniform throughout the research area [26]. The OLS model was employed to diagnose and identify the most reliable determinants of stunting for the Geographically Weighted Regression (GWR) analysis [27].

Before proceeding to the local model, the six assumptions of the OLS model (the explanatory variables should have the expected relationship, the significance of each explanatory variable, the randomness of residuals, assuring the statistical no significance of Jarque-Bera statistics, Variance Inflation Factor (VIF) value, and the strength of R-square) were checked. The multicollinearity was determined using the VIF values. Predictors with VIF values $> 10$, i.e., the cut point for declaring the presence of multicollinearity, were not found in this data. After confirming the OLS model's assumptions, the local model, i.e., GWR were used to describe geographically varying relationships, assuming that the relationship between variables varies geographically [28].

**Geographically weighted regression (GWR).** A local geographic statistical method that considers relationships between the dependent and independent variables to be non-stationary or heterogeneous across EAs [22, 27, 29]. The GWR analysis is used when the Koenker

statistics show significance (p-value of 0.05), indicating that the associations between the dependent and independent variables differ by geographical location. The explanatory variable coefficients in the GWR analysis have varying values throughout the study's area. The adjusted R-squared and corrected Akaike Information Criteria (AICc) were used to compare the OLS (global model) and GWR (local model) models. The best-fitted model for the data was the one with the lowest AICc value and the highest adjusted R-squared value.

**Ethical consideration.** Data access permission was received from the measure DHS program via an online request at http://www.dhsprogram.com. The data used in this research were publicly available and did not contain any personal identifiers.

## Result

### Descriptive results

A total weighted sample of 5,490 children under the age of five were included in this study. More than half (51.51%) of them were males. Two third (62.80%) of them were between the age of 24–59 months, and the vast majority (75.19%) were form rural residencies. Half of (50.50%) of the mother's age were between 20–29 years and 53.46% of the mothers do not attend a formal education. Regarding breastfeeding, 6.60% of them were never breastfed (Table 1).

### The prevalence of stunting among under-five children in Ethiopia

The prevalence of stunting among under five children was 33.58% (95%CI: 32.34%, 34.84). The lowest prevalence of stunting among under five children was observed in Addis Ababa (12.28%), Gambela (15.74%), and Dire Dawa (21.95%) regions. On the contrary, highest prevalence of stunting was observed in Tigray (46.95%), Afar (38.08%) and Amhara (37.80%) regions (Fig 1).

**Factors associated with stunting among under-five children.** *Random effect analysis.* The ICC in the empty model was 12.01% (95%CI: 9.13%, 15.39%), indicating that 12.01% of the total variability for stunting among under-five children was due to differences between clusters/ EA, with the remaining unexplained 87.99% that is attributed by individual differences. Furthermore, based on the Log-likelihood ratio test with a result of $X^2 = 177.67$ and a p-value of 0.001, it is recommended that we use a mixed-effect model instead of the basic model. The models were compared with deviance (-2Log-likelihood Ratio (-2LLR)) and the mixed effect robust Poisson regression model best fitted the data since it had the lowest deviance value.

*Fixed effect analysis.* Maternal age, maternal educational status, household income index, sex of the child, age of the child, area, and community maternal illiteracy were all significantly associated with stunting among under-five children in a multivariable mixed-effect Poisson regression with robust variance analysis.

The prevalence of stunting among children whose mother's age is >40 decreased by 26% (APR = 0.74, 95%CI: 0.55, 0.99). Compared to children whose mothers do not have a formal education the prevalence of stunting were decreased by 26% (APR = 0.74, 95%CI: 0.60, 0.91) and 39% (APR = 0.61, 95%CI: 0.44, 0.84) among children whose mother had secondary and higher educational status, respectively. Children in the rich household wealth index had the prevalence of stunting decreased by 13% (APR = 0.87, 95%CI: 0.76, 0.99) compared to children from poor household wealth index. The prevalence of stunting among children aged 6–23 months were 1.87 times (APR = 1.87, 95%CI: 1.53, 2.28) higher compared to children aged 0–5 months (Table 2).

**Table 1. Socio-demographic characteristics of the under-five children in Ethiopia, 2019 (n = 5,490).**

| Variables | Weighted frequency | Percentage |
|---|---|---|
| **Individual level variables** | | |
| **Maternal age** | | |
| <20 | 260 | 4.74 |
| 20–29 | 2,773 | 50.50 |
| 30–39 | 1,981 | 36.09 |
| >40 | 476 | 8.67 |
| **Maternal educational status** | | |
| No formal education | 2,935 | 53.46 |
| Primary | 1,948 | 35.49 |
| Secondary | 413 | 7.52 |
| Higher | 194 | 3.53 |
| **Household wealth index** | | |
| Poor | 2,494 | 45.42 |
| Middle | 1,039 | 18.92 |
| Rich | 1,958 | 35.66 |
| **Family size** | | |
| 1–4 | 1,633 | 29.74 |
| 5–8 | 3,117 | 56.77 |
| >8 | 741 | 13.49 |
| **Sex of household head** | | |
| Male | 4,747 | 86.46 |
| Female | 743 | 13.54 |
| **Source of drinking water** | | |
| Not improved | 1,881 | 34.26 |
| Improved | 3,610 | 65.74 |
| **Child related characteristics and maternal health care utilizations** | | |
| **Sex of the child** | | |
| Male | 2,828 | 51.51 |
| Female | 2,662 | 48.49 |
| **Age of the child (In months)** | | |
| 0–5 | 552 | 10.05 |
| 6–23 | 1,491 | 27.15 |
| 24–59 | 3,448 | 62.80 |
| **Number of under-five children** | | |
| 1 | 2,492 | 45.40 |
| 2 | 2,488 | 45.31 |
| ≥3 | 510 | 9.29 |
| **Place of delivery** | | |
| Home | 2,823 | 51.41 |
| Health facilities | 2,667 | 48.59 |
| **Type of birth** | | |
| Single | 5,345 | 97.35 |
| Multiple | 146 | 2.65 |
| **Birth order** | | |
| 1–3 | 2,957 | 53.86 |
| 4–6 | 1,705 | 31.06 |
| >6 | 828 | 15.08 |

(*Continued*)

**Table 1.** (Continued)

| Variables | Weighted frequency | Percentage |
|---|---|---|
| **Birth interval (In months)** | | |
| < 24 | 966 | 17.60 |
| ≥ 24 | 4,524 | 82.40 |
| **Duration of breastfeeding** | | |
| Ever breastfed, not currently breastfed | 2,803 | 51.06 |
| Never breastfed | 362 | 6.60 |
| Still breastfeeding | 2,325 | 42.34 |
| **ANC visit** | | |
| No | 997 | 18.16 |
| 1–3 | 1,217 | 22.17 |
| ≥4 | 3,276 | 59.67 |
| **Community level variables** | | |
| **Residence** | | |
| Urban | 1,362 | 24.81 |
| Rural | 4,128 | 75.19 |
| **Region** | | |
| Tigray | 371 | 6.75 |
| Afar | 85 | 1.54 |
| Amhara | 1,046 | 19.05 |
| Oromia | 2,196 | 39.99 |
| Somalia | 407 | 7.41 |
| Benishangul | 65 | 1.19 |
| SNNPR* | 1,094 | 19.93 |
| Gambela | 25 | 0.45 |
| Harari | 16 | 0.30 |
| Addis Ababa | 156 | 2.85 |
| Dire Dawa | 30 | 0.54 |
| **Community maternal illiteracy level** | | |
| Low | 2,714 | 49.43 |
| High | 2,776 | 50.57 |

* SNNPR: southern nation nationality and people's region

**Spatial distribution of stunting among under-five children.** Tigray, Amhara, Afar, Benishangul Gumuz, SNNPR, and portions of western Oromia had the highest prevalence of stunting among children under the age of five (Fig 2). Significant spatial variation in stunting among under-five children was observed across the country with a global Moran's I value of 0.400 (p-value0.01) (Fig 3). The statistically significant hotspot areas of stunting were identified in the west and south Afar, Tigray, Amhara and east SNNPR regions. Significant cold spot areas were observed in Gambela, Addis Ababa, northwest Oromia, Dire Dawa and Harari regions (Fig 4).

**The global ordinary least square regression analysis.** Exploratory Regression was used to test all possible combinations of the input candidate explanatory variables to find OLS models that best explain the dependent variable. With AICc = -265.46, the model explained about 25% (adjusted $R^2$ = 0.25) of the variation in stunting among children under the age of five in the ordinary least square analysis. The model was statistically significant, as shown by the significant Joint F-statistics and Wald statistics ($p < 0.05$). The statistically significant test of the

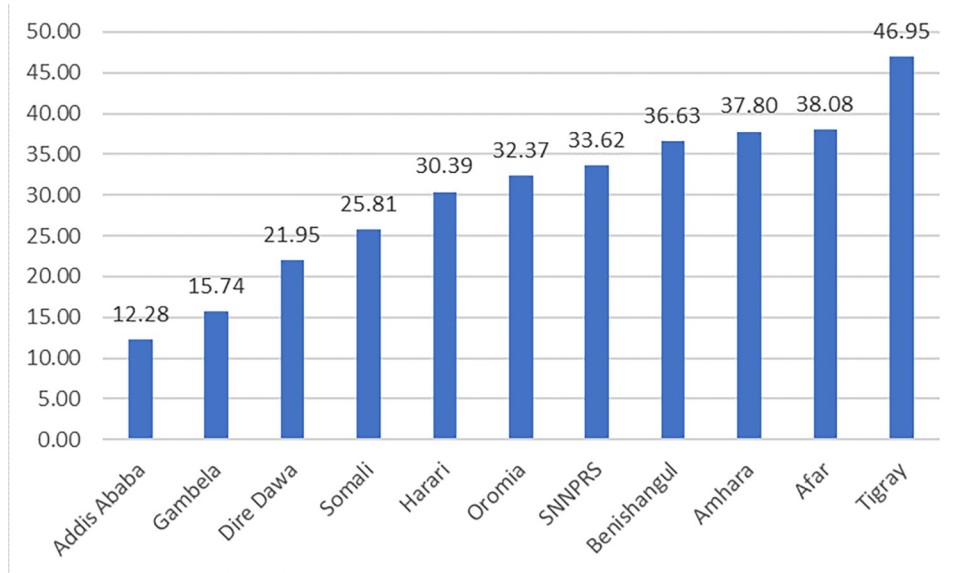

**Fig 1. The prevalence of stunting among under five children across regions in Ethiopia, 2019.**

Koenker statistics indicates there is a spatial heterogeneity or non-stationarity in the relationship between dependent and independent variables across the study area. This suggests that GWR should be used (because the Koenker statistics proved the relationship's non-stationarity) given that it presumes that the relationship between independent and dependent variables is spatially heterogeneous or vary across space. The proportion of mothers who do not have formal education, the proportion of children aged 6–23 months, the proportion of women who reside in poor households, and the proportion of male headed households were significantly associated with the percentage of stunting among under-five children in the OLS model (Table 3).

**Geographically weighted regression analysis.** The GWR analysis indicated a considerable improvement over the global model (OLS). The AICc value in the GWR model decreased from -265.46 in the OLS model to -296.37. The difference suggested that the GWR model best describes the spatial heterogeneity of stunting among children under the age of five. Furthermore, the adjusted $R^2$ in the GWR model was 0.37, the model's capacity to explain stunting among children under the age of five has been increased by employing GWR. This suggests that GWR improved the model's explanatory power of the OLS model by roughly 12% (Table 4).

The proportion of stunted under five children increased with the proportion of mothers without any formal education. As the proportion of mothers with no formal education increased, the percentage of stunted children under five also increased in the entire Dire Dawa, Harari, eastern Afar and Somali. The geographic area with the highest coefficient of the proportion of mothers who had no formal education has colored in red points (Fig 5). In Tigray, Amhara, SNNPR and Addis Ababa, the proportion of mothers with poor household wealth status was found to be strongly and positively associated with an increased risk of stunting in children under the age of five (Fig 6). Stunting among under five children was positively and significantly associated with the proportion of children aged 6–23 months (Fig 7). The proportion of children living in male-headed households was found to have a positive association with an increased risk of stunting in SNNPR, Somali, Harari, Dire Dawa, central Oromia and Addis Ababa while negatively associated in Benishangul Gumuz (Fig 8).

**Table 2. Bi-variable and multi-variable mixed effect robust Poisson regression analysis of stunting among under-five children in Ethiopia, 2019.**

| Variables | Stunted | | CPR | APR |
|---|---|---|---|---|
| | No | Yes | | |
| **Individual level variables** | | | | |
| **Maternal age** | | | | |
| <20 | 184 | 76 | 1 | 1 |
| 20–29 | 1,864 | 908 | 1.15 (0.94, 1.12) | 1.05 (0.85, 1.28) |
| 30–39 | 1,247 | 734 | 1.28 (1.05, 1.57) | 1.04 (0.83, 1.30) |
| >40 | 351 | 125 | 0.99 (0.76, 1.30) | 0.74 (0.55, 0.99)* |
| **Maternal educational status** | | | | |
| No formal education | 1,816 | 1,119 | 1 | 1 |
| Primary | 1,326 | 622 | 0.87 (0.79, 0.95) | 0.99 (0.89, 1.09) |
| Secondary | 341 | 72 | 0.57 (0.46, 0.69) | 0.74 (0.60, 0.91)* |
| Higher | 163 | 31 | 0.42 (0.30, 0.57) | 0.61 (0.44, 0.84)* |
| **Household wealth index** | | | | |
| Poor | 1,575 | 919 | 1 | 1 |
| Middle | 641 | 399 | 1.02 (0.91, 1.14) | 1.00 (0.89, 1.13) |
| Rich | 1,431 | 527 | 0.69 (0.61, 0.79) | 0.87 (0.76, 0.99)* |
| **Sex of household head** | | | | |
| Male | 3,096 | 1,651 | 1 | 1 |
| Female | 551 | 193 | 0.81 (0.72, 0.92) | 0.89 (0.79, 1.01) |
| **Source of drinking water** | | | | |
| Not improved | 1,230 | 651 | 1 | 1 |
| Improved | 2,416 | 1,193 | 0.87 (0.79, 0.97) | 0.99 (0.90, 1.09) |
| **Child-related characteristics and maternal health care utilization** | | | | |
| **Sex of the child** | | | | |
| Male | 1,808 | 1,020 | 1 | 1 |
| Female | 1,838 | 824 | 0.92 (0.85, 0.99) | 0.91 (0.85, 0.98)* |
| **Age of the child (In months)** | | | | |
| 0–5 | 462 | 90 | 1 | 1 |
| 6–23 | 1,027 | 463 | 1.85 (1.51, 2.26) | 1.87 (1.53, 2.28)* |
| 24–59 | 2,158 | 1,290 | 2.19 (1.81, 2.64) | 2.16 (1.79, 2.61)* |
| **Birth order** | | | | |
| 1–3 | 2,017 | 940 | 1 | 1 |
| 4–6 | 1,085 | 620 | 1.19 (1.09, 1.29) | 1.09 (0.98, 1.21) |
| >6 | 545 | 284 | 1.09 (0.96, 1.23) | 1.11 (0.96, 1.29) |
| **Birth interval (In months)** | | | | |
| < 24 | 619 | 347 | 1 | 1 |
| ≥ 24 | 3,027 | 1,497 | 0.88 (0.79, 0.97) | 0.94 (0.85, 1.04) |
| **Community level variables** | | | | |
| **Residence** | | | | |
| Urban | 1,041 | 322 | 1 | 1 |
| Rural | 2,606 | 1,522 | 1.70 (1.44, 2.01) | 1.08 (0.89, 1.30) |
| **Region** | | | | |
| Addis Ababa | 137 | 19 | 1 | 1 |
| Tigray | 197 | 174 | 3.83 (2.55, 5.78) | 2.68 (1.78, 4.03)* |
| Afar | 52 | 32 | 3.03 (2.03, 4.52) | 1.82 (1.19, 2.77)* |
| Amhara | 651 | 395 | 3.13 (2.09, 4.69) | 1.97 (1.29, 2.99)* |

*(Continued)*

**Table 2.** (Continued)

| Variables | Stunted | | CPR | APR |
|---|---|---|---|---|
| | **No** | **Yes** | | |
| Oromia | 1,485 | 711 | 2.61 (1.75, 3.90) | 1.68 (1.11, 2.55)* |
| Somalia | 302 | 105 | 2.06 (1.33, 3.18) | 1.18 (0.75, 1.86) |
| Benishangul | 42 | 24 | 2.83 (1.85, 4.32) | 1.82 (1.15, 2.87)* |
| SNNPR | 726 | 368 | 2.72 (1.81, 4.09) | 1.73 (1.13, 2.65)* |
| Gambela | 21 | 4 | 1.32 (0.85, 2.06) | 0.97 (0.61, 1.51) |
| Harari | 11 | 5 | 2.25 (1.44, 3.52) | 1.69 (1.12, 2.56)* |
| Dire Dawa | 23 | 7 | 1.83 (1.18, 2.83) | 1.32 (0.87, 1.99) |
| **Community maternal illiteracy level** | | | | |
| Low | 1,933 | 780 | 1 | 1 |
| High | 1,713 | 1,064 | 1.47 (1.29, 1.66) | 1.17 (1.04, 1.32)* |

*p-value <0.05

CPR: Crude prevalence ratio

APR: Adjusted prevalence ratio

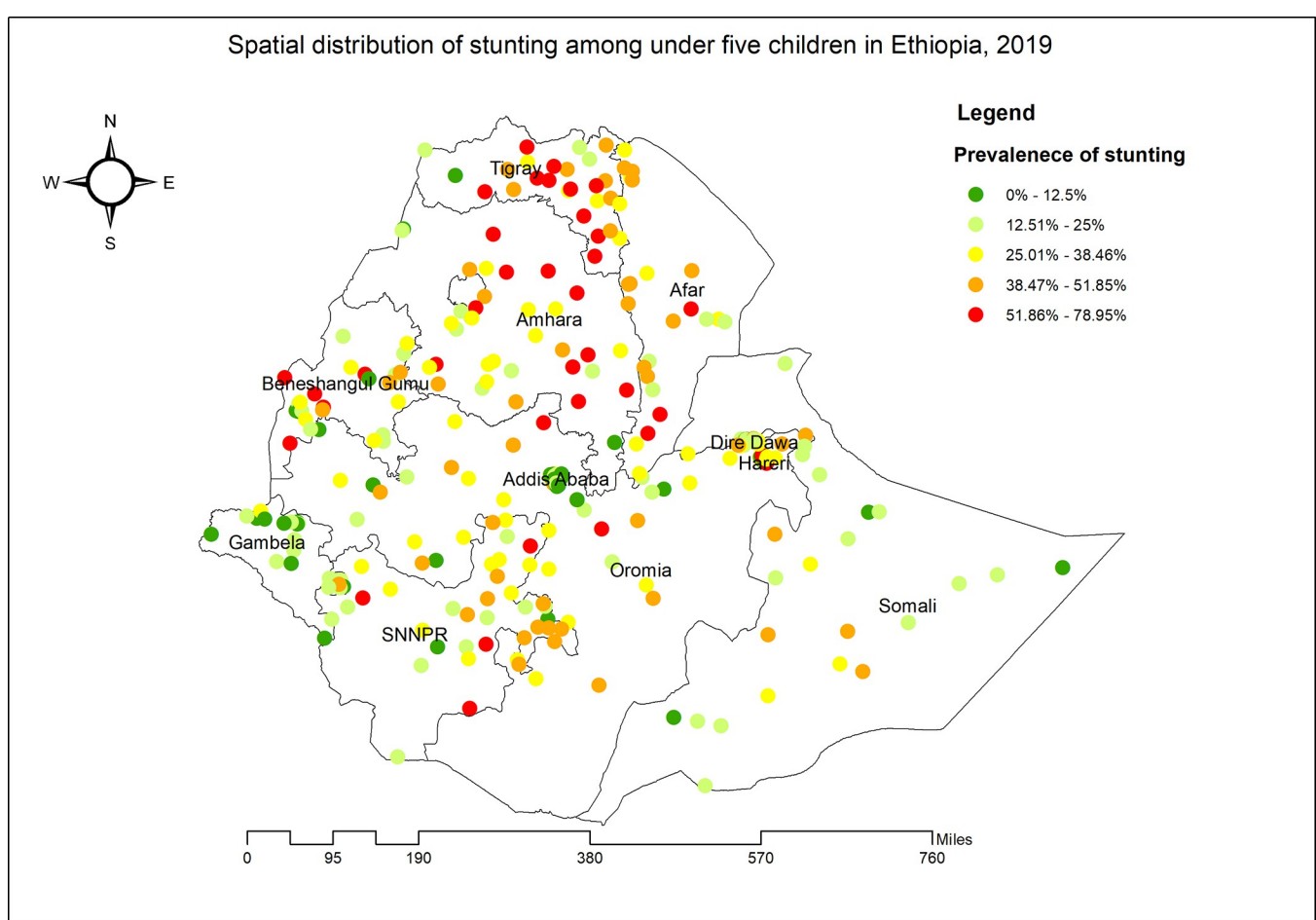

**Fig 2. Spatial distribution of stunting among under-five children in Ethiopia, 2019.**

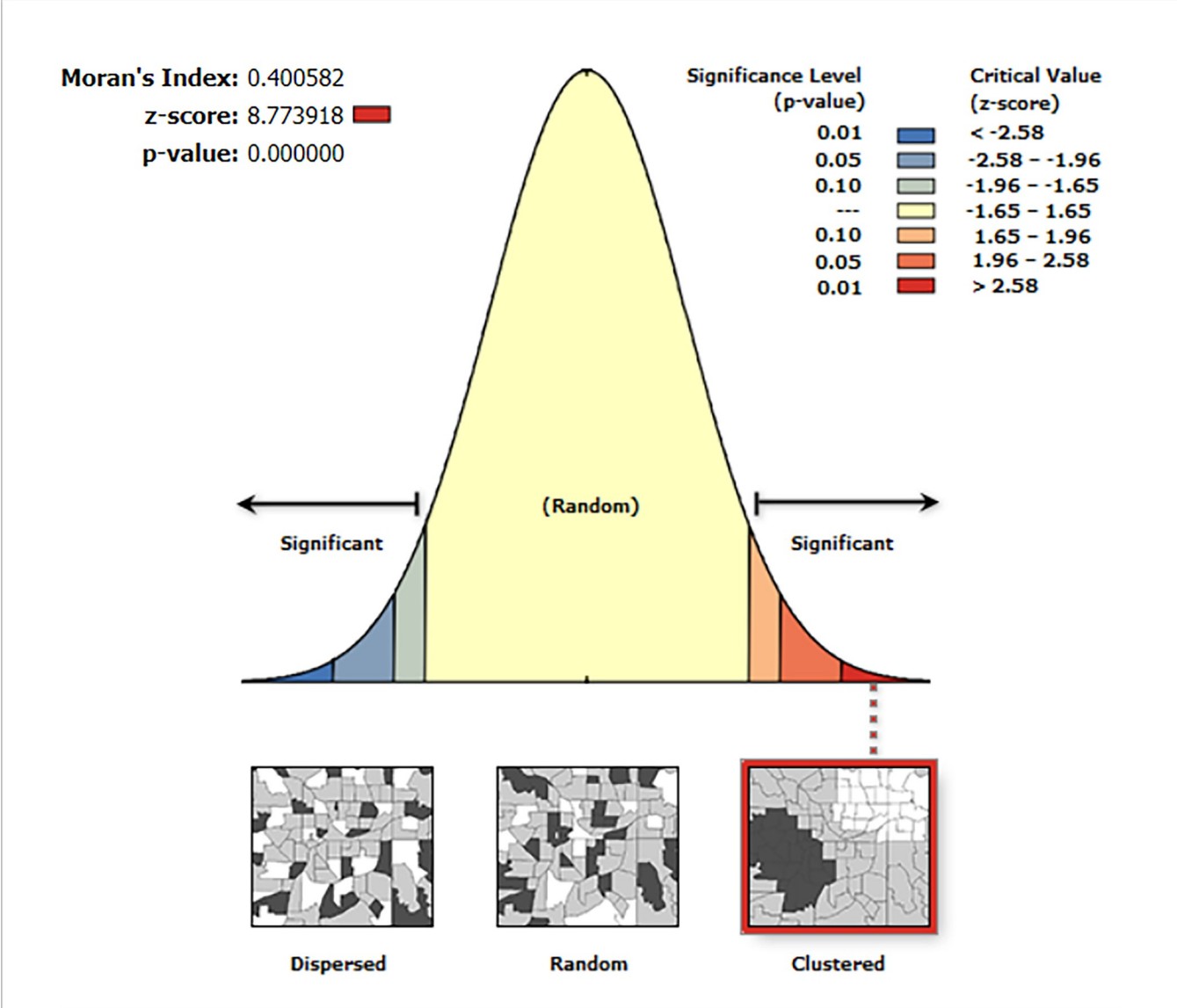

**Fig 3. Global spatial autocorrelation analysis of stunting among under five-children in Ethiopia, 2019.**

## Discussion

In this study, the prevalence of stunting among children under the age of five in Ethiopia was 33.58% (95%CI: 32.34%, 34.84). The highest prevalence of stunting was observed in Tigray (46.95%) while the lowest prevalence was in Addis Ababa (12.28%). The prevalence of stunting is consistent with studies conducted in SSA countries [8, 30, 31]. Most SSA countries specifically, East African nations share many socioeconomic and health-related characteristics, and this could be one of the reasons for the consistent countrywide prevalence of stunting. Nevertheless, this finding was higher than the prevalence of stunting reported by a study done in

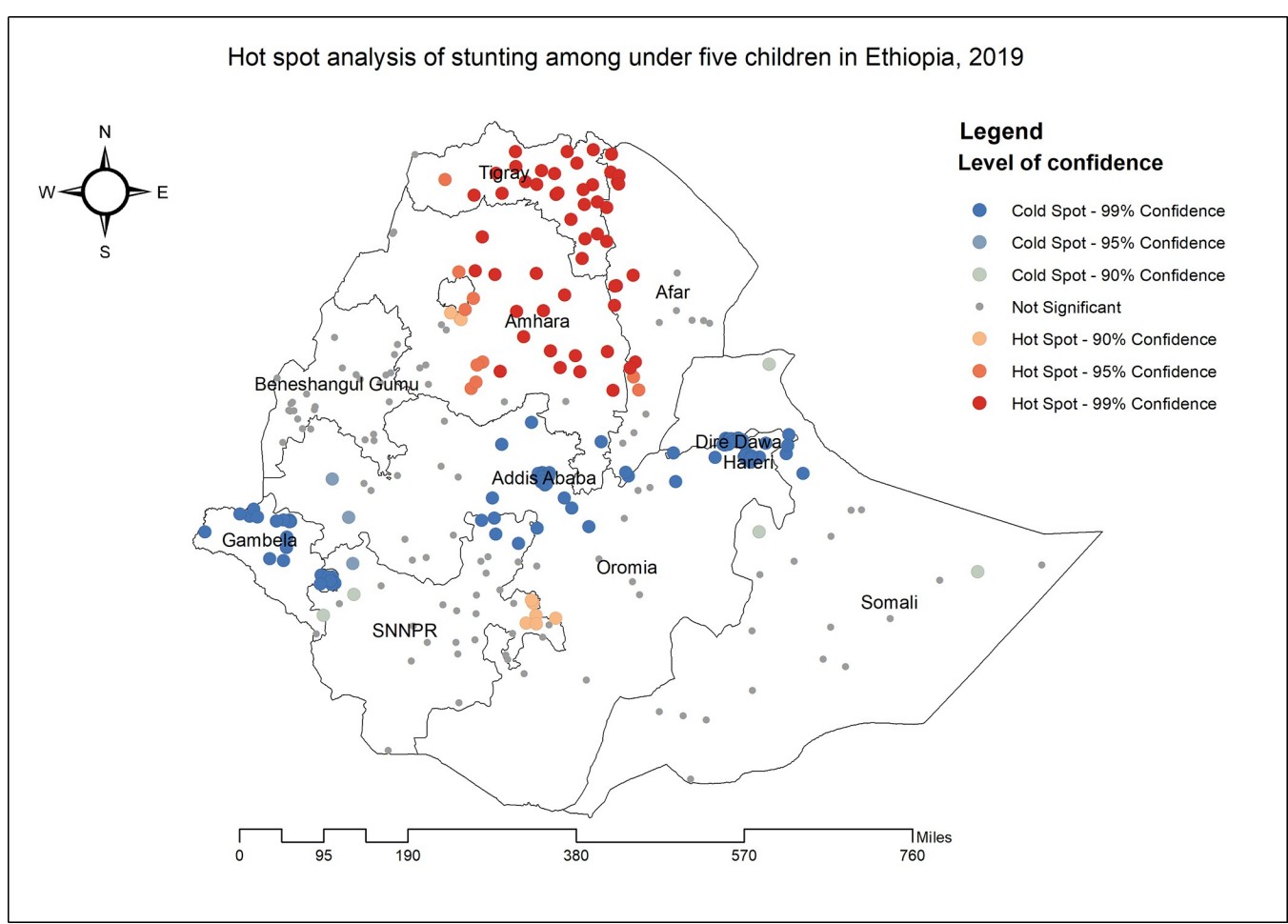

**Fig 4. The Getis Ord Gi statistical analysis of hot spots of stunting among under-five children in Ethiopia, 2019.**

**Table 3. Ordinary least square (OLS) regression analysis.**

| Variable | Coefficient | Robust std-error | Robust t-statistics | Robust p-value (non-stationarity p-value) | VIF |
|---|---|---|---|---|---|
| Intercept | 0.12 | 0.03 | 3.91 | 0.00012 | |
| Proportion of mothers aged < 20 years | 0.03 | 0.10 | 0.04 | 0.97 | 1.13 |
| Proportion of mothers who had no formal education | 0.24 | 0.04 | 5.61 | 0.000 | 1.95 |
| Proportion of mother with poor household wealth status | 0.07 | 0.03 | 2.08 | 0.03 | 2.00 |
| Proportion of children aged 6–23 months | 0.19 | 0.08 | 2.42 | 0.02 | 1.07 |
| Proportion of male household heads | 0.14 | 0.04 | 3.84 | 0.000 | 1.09 |
| Proportion of birth order >6 | 0.05 | 0.09 | 0.55 | 0.58 | 1.39 |
| **Ordinary least square regression diagnostic** | | | | | |
| Number of observation | | 305 | | | |
| Joint F-statistics | | 16.06: p-value <0.01 | | | |
| Joint Wald statistics | | 117.93: p-value <0.01 | | | |
| Koenker (BP) statistics | | 19.19: p-value <0.01 | | | |
| Jarque- Bera | | 5.79: p-value = 0.06 | | | |

VIF: Variance Inflation Factor

**Table 4. Model comparison of OLS and GWR model.**

| Model comparison parameter | OLS model | GWR model |
|---|---|---|
| AICc | -265.46 | -296.37 |
| Adjusted R-squared | 0.25 | 0.37 |

china [32] and Bangladesh [33]. The dependence of Ethiopians on agriculture is greatly affected by weather and climate factors, such as temperature, precipitation, light, and extreme events. This may be the reason for their susceptibility to food insecurity. Furthermore, their limited ability to adapt exacerbates the situation [34, 35]. Moreover, there is a significant association between poverty and stunted growth in children below the age of five [36, 37]. Asian nations like china and Bangladesh is more affluent and has better access to essential maternal and child health care services than Ethiopia, therefore this may be an explanation why stunting is not prevalent in those nations.

This study showed that the spatial distribution of stunting among children under the age of five in Ethiopia were clustered and the hotspots of stunting were identified in the entire Tigray, central, northeast and southern Amhara, north, west and south Afar and east SNNPR regions. Here are some potential local and regional factors that could contribute to these disparities: Tigray and Central Amhara is predominantly agricultural [38]. Traditional farming practices,

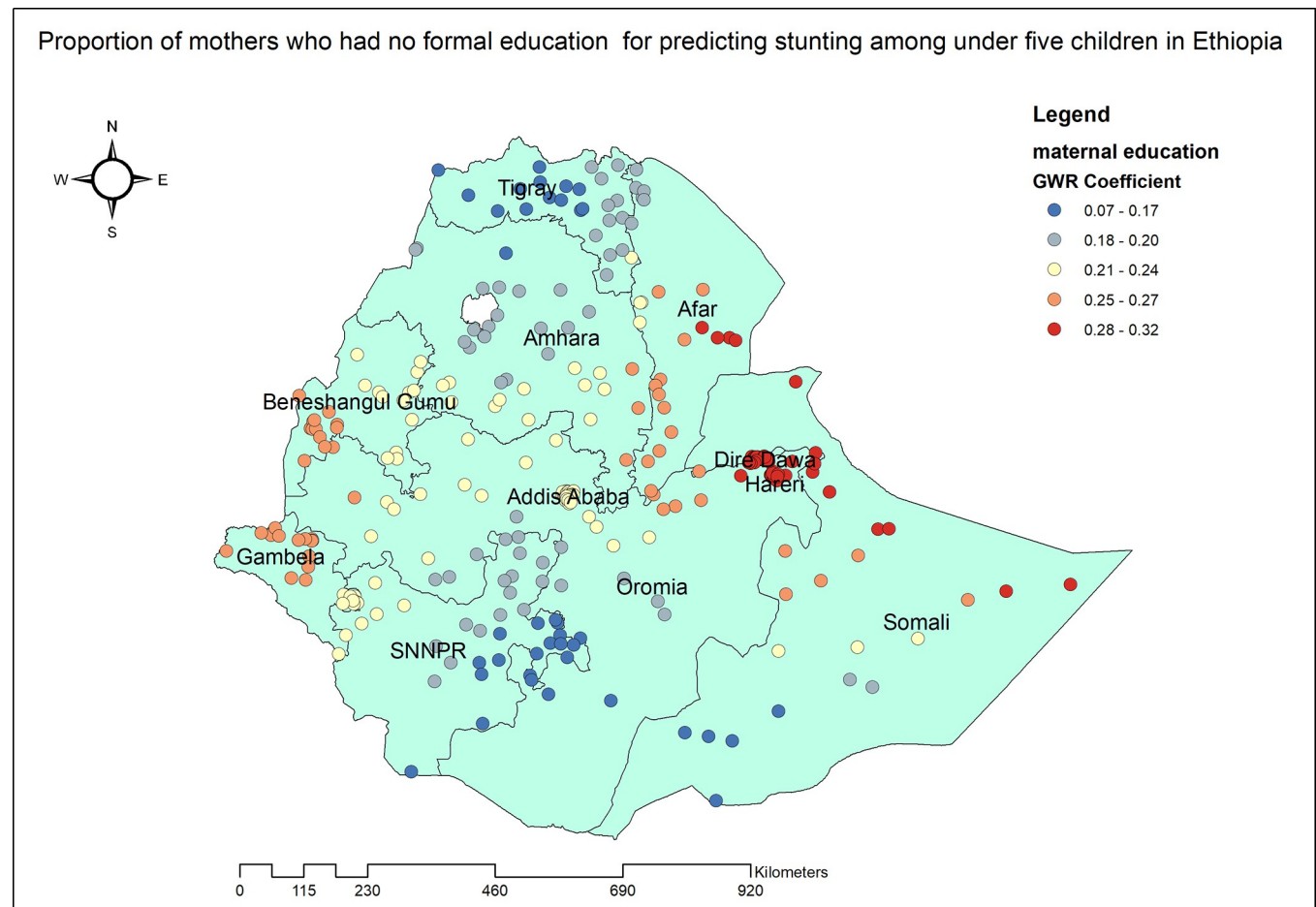

**Fig 5. Mothers who had no formal education GWR coefficients for predicting stunting among children under the age of five in Ethiopia, 2019.**

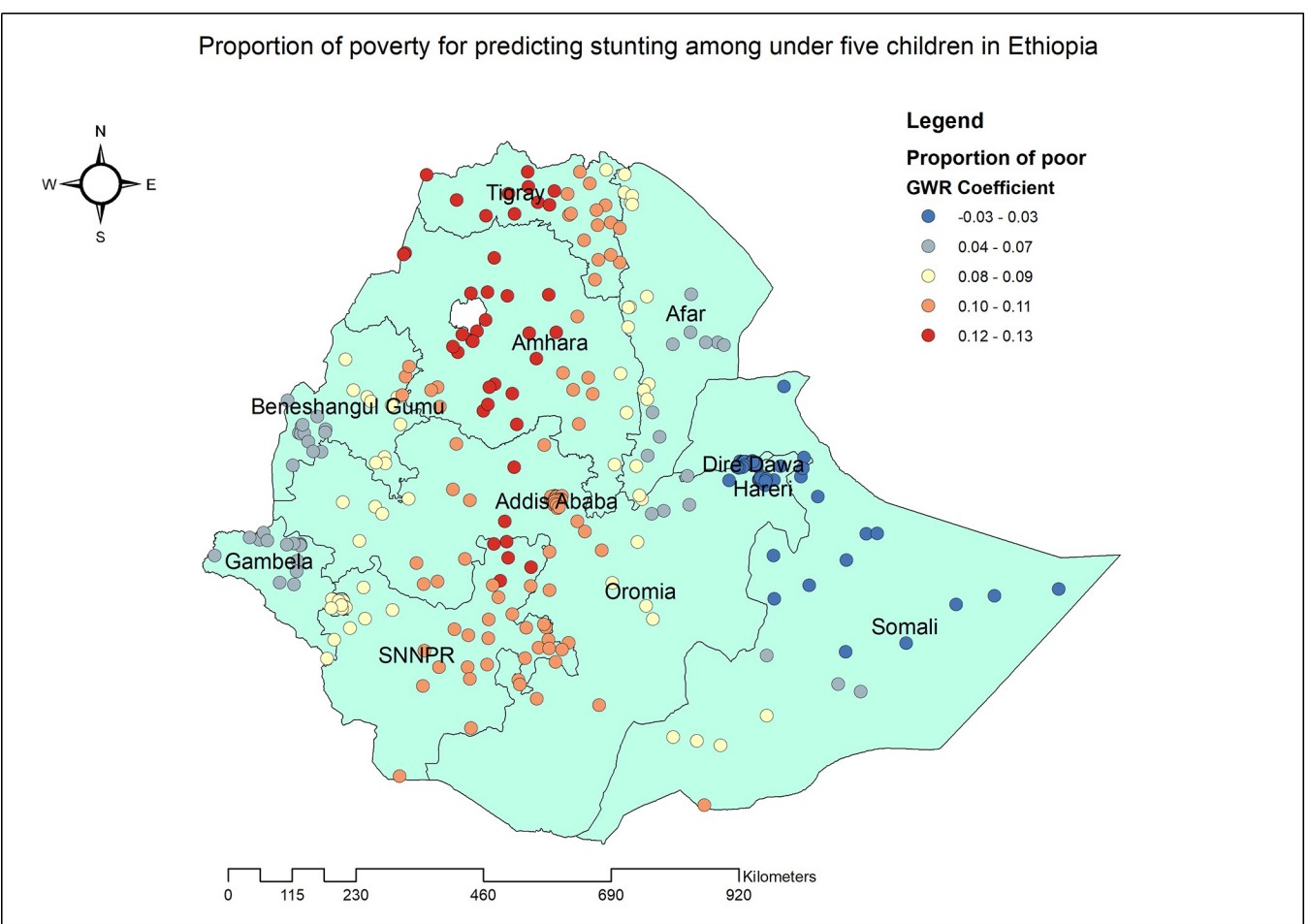

**Fig 6. Children who resides in household with poor wealth index GWR coefficients for predicting stunting among children under the age of five in Ethiopia, 2019.**

limited crop diversity, and inadequate access to nutritious foods can contribute to stunting [39]. This region faces recurrent food shortages due to erratic rainfall and poor agricultural productivity [40]. Remote areas in northeast Amhara may have limited access to healthcare services, affecting child health [41]. Afar is a desert region with extreme temperatures and limited vegetation [42]. Malnutrition due to food scarcity and water shortages is common [43]. Nomadic pastoralism makes it challenging to access healthcare and nutritious foods consistently and Sparse health facilities hinder timely interventions [44]. West Afar faces recurrent droughts and famines, leading to food insecurity and malnutrition [45]. SNNPR is ethnically diverse, with varying dietary practices. Some communities may lack awareness of balanced nutrition [40]. These disparities arise from a complex interplay of socio-economic, cultural, environmental, and infrastructural factors.

Even though, policies and programs have been implemented to reduce stunting among under five children like strengthening rural agriculture in order to promote food security; Health system decentralization, including health extension workers to enhance rural access to health services; and varied poverty-reduction strategies [46], there was significant spatial heterogeneity in stunting throughout the study area. Inadequate dietary intake of nutrients [47] and a high burden of an infectious disease as a result of repeated exposure to poor hygiene and

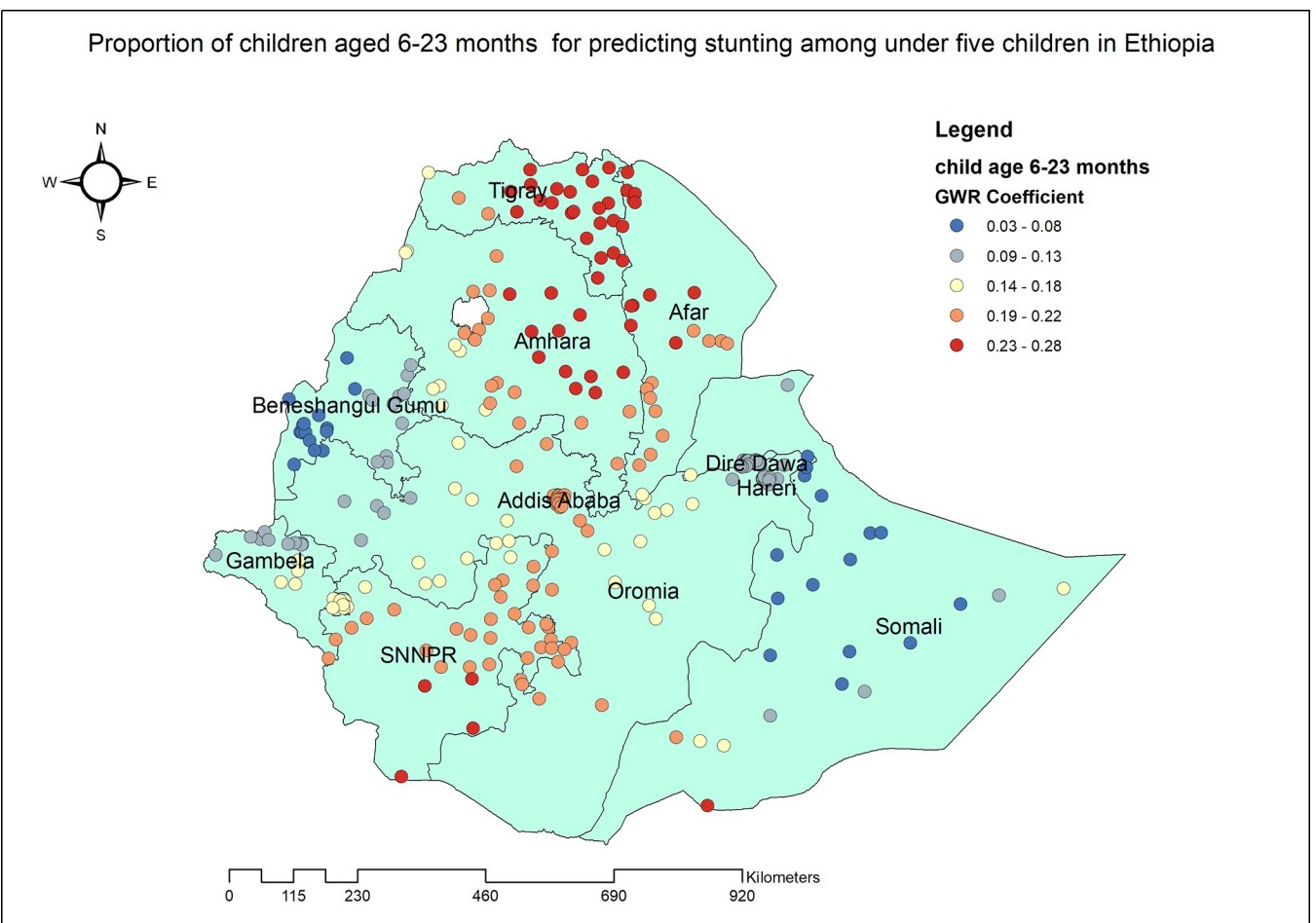

**Fig 7. Children aged 6–23 months GWR coefficients for predicting stunting among children under the age of five in Ethiopia, 2019.**

environmental factors that promote parasite transmission and spread, which are direct causes of stunting [4, 48] are probable explanations.

In the geographically weighted regression, the maternal educational status, poverty, child age, and male-headed household significantly predicted spatial variation of stunting among children under the age of five. There is a positive relationship between being a child of mother who had no formal education and stunting in the entire Dire Dawa, Harari, eastern Afar and Somali. This is supported by a study conducted in India [49] and Ethiopia [20] which found that the mother's educational level is a significant spatial predictor of child malnutrition. Decades of research have shown that a mother's education has a significant impact on the nutritional status of her children. Maternal education improves the ability to acquire health knowledge, follow recommended feeding practices, and gain more control over resources, all of which contribute to better child health outcomes [50–54]. It is unequivocal that mothers without formal education exhibit a significantly lower level of understanding and awareness regarding the importance of proper nutrition during pregnancy and lactation, leading to an increased likelihood of stunted growth among their children [55]. In addition, unlike illiterate mothers, literate mothers are economically empowered, allowing them to comply with recommended infant and young child feeding practices [56] and provide more diverse meals to their children [57].

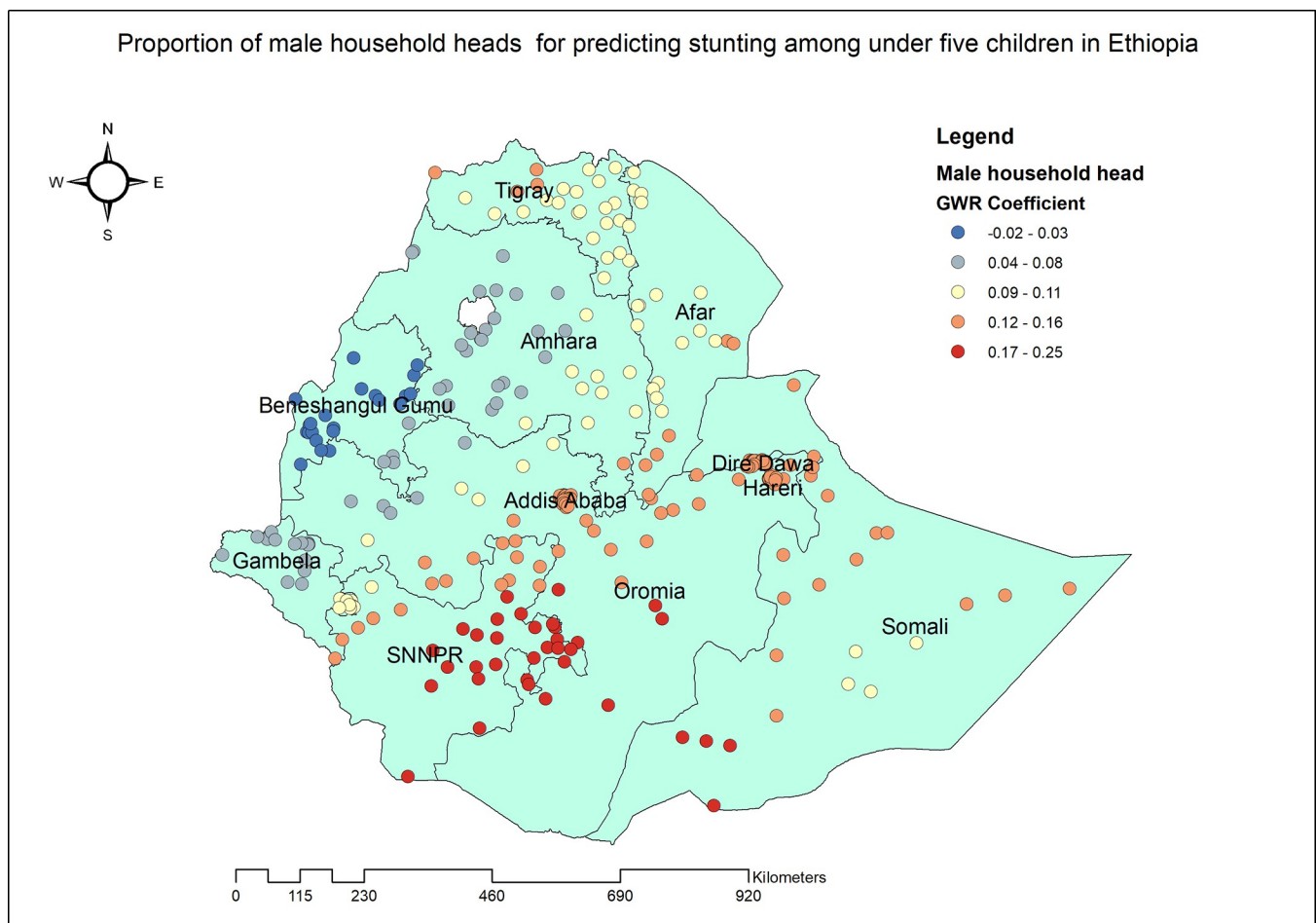

**Fig 8. Children living in male-headed households GWR coefficients for predicting stunting among children under the age of five in Ethiopia, 2019.**

In Tigray, Amhara, SNNPR regions and Addis Ababa, poverty and stunting were positively and strongly associated. A Bayesian geostatistical analysis done in Ethiopia found that wealth status as a significant predictor of stunting among under-five children [13]. The underlying reasons for the increased prevalence of growth failure in children from low-income households can be attributed to insufficient food intake, a higher likelihood of illness, and limited access to essential healthcare services [58, 59]. It is widely acknowledged that children from low-income households are at a greater risk of growth failure due to inadequate nutrition, which has a direct impact on their overall development and well-being. In addition, these children are more likely to suffer from illnesses, which further exacerbates their growth failure. Moreover, the limited access to essential healthcare services adds an additional layer of complexity to the issue, as early intervention and treatment can significantly improve the chances of recovery [60].

Poverty, despite being a risk factor for childhood stunting, found to paradoxically reduce the risk of stunting in certain contexts. This could be due Community programs and public health interventions, particularly those aimed at impoverished communities, play a crucial role in addressing stunting. These efforts often prioritize the provision of varied and nutritious foods, significantly contributing to the fight against childhood stunting. Furthermore Social safety net programs, including initiatives like Poverty Reduction Support, play a crucial role in

supporting families in need. These programs have a direct impact on child nutritional status by addressing the underlying determinants of nutrition [61–67]. In brief, Ethiopia's diverse strategies, regional distinctions, cost considerations, and community-driven initiatives play a role in producing different results. Ongoing research, adaptability, and collaboration are essential for maintaining positive advancements in nutrition and health outcomes nationwide.

Our study reports a strong association between stunting and the percentage of children aged 6–23 months. Notably, this finding aligns with previous research conducted in Rwanda [68], Ethiopia [69], and Zimbabwe [70]. The age group of young children may necessitate continuous feedings for optimal growth and development. However, the introduction of complementary foods or the interruption of breastfeeding can lead to a decrease in the energy and nutritional density of their diet, thereby posing a threat to their overall well-being. It is imperative to ensure that the nutritional requirements of this age group are met to promote their healthy growth and development. In addition, without breastmilk and its immunological components, the child does not have the same level of protection against illnesses [33].

In SNNPR, Somali, Harari, Dire Dawa, central Oromia, and Addis Ababa, the proportion of children living in male-headed households was found to have a positive association with a higher likelihood of stunting. Women-headed households tend to provide more resources for their children than men-headed households [71] and make better antenatal care choices [72] which in turn have a positive outcome on the child's nutritional status. However, in some part of the country male-headed households were associated with decreased likelihood of stunting. It is imperative to consider gender dimensions when addressing child nutrition and health outcomes, as the relationship between household gender dynamics and child stunting is highly complex [73].

This study contributes to the existing comprehension of the implications of geographical variations on stunting throughout the entire nation. Furthermore, assists in identifying different hotspot locations throughout the country and the actual effects of variables on stunting in each particular geographical location, which is critical for prevention and intervention. Addressing stunting requires region-specific interventions, including improved healthcare, sanitation, education, and community engagement.

## Strength and limitation of the study

This study used nationwide representative data, which improves generalizability. The outcomes of the spatial and geographic weighted regression analyses can help policymakers and program planners develop spatially focused public health measures to decrease the incidence of stunting. The study has the following limitations, because this study used data from a cross-sectional study design, declaring causality between the dependent variable and the explanatory variables was difficult. Furthermore, for the purpose of privacy, the geographic positions (GPs) of enumeration areas were displaced up to 2 kilometers in urban areas, 5 kilometers for most enumeration areas in rural areas, and 10 kilometers for 1% of clusters in rural areas, which might affect the estimated cluster effects in the spatial regression.

## Conclusion

In Ethiopia, under-five children suffering from stunting have been found to exhibit a spatially clustered pattern. This implies that the distribution of stunting is not uniform but rather concentrated in specific geographic regions. Such findings can aid in formulating targeted interventions to effectively address the issue of stunting in the country. By directing resources and efforts towards these high-risk areas, significant progress can be achieved in improving the health and overall well-being of affected children. The prevalence of stunting among children

under the age of five is particularly high in geographical areas characterized by a high proportion of male-headed households, a high proportion of 6–23 months of children, high levels of poverty, and a high proportion of maternal illiteracy. The mitigation of stunting in Ethiopia necessitates the implementation of targeted public health interventions that are focused on high-risk groups and specific geographic regions. Such interventions have the potential to significantly reduce the incidence of stunting among the most vulnerable populations in the country. In order to address this issue, it is imperative that a collaborative approach is taken between governmental and non-governmental entities. This approach will ensure the design and implementation of policies and programs that target the contributing factors and prevent the transmission of stunting from one generation to the next. These efforts must prioritize the health and wellbeing of children, as well as their families and communities, and must be sustained over time.

## Acknowledgments

We would like to thank the measure DHS program for providing the datasets.

## Author Contributions

**Conceptualization:** Beminate Lemma Seifu, Getayeneh Antehunegn Tesema.

**Data curation:** Beminate Lemma Seifu, Getayeneh Antehunegn Tesema.

**Formal analysis:** Beminate Lemma Seifu, Getayeneh Antehunegn Tesema.

**Investigation:** Tirualem Zeleke Yehuala.

**Methodology:** Beminate Lemma Seifu, Bezawit Melak Fentie, Tirualem Zeleke Yehuala, Abdulkerim Hassen Moloro, Kusse Urmale Mare.

**Software:** Beminate Lemma Seifu, Getayeneh Antehunegn Tesema, Bezawit Melak Fentie, Abdulkerim Hassen Moloro, Kusse Urmale Mare.

**Supervision:** Getayeneh Antehunegn Tesema, Abdulkerim Hassen Moloro, Kusse Urmale Mare.

**Validation:** Beminate Lemma Seifu, Getayeneh Antehunegn Tesema, Tirualem Zeleke Yehuala, Abdulkerim Hassen Moloro, Kusse Urmale Mare.

**Visualization:** Beminate Lemma Seifu, Bezawit Melak Fentie, Tirualem Zeleke Yehuala, Abdulkerim Hassen Moloro, Kusse Urmale Mare.

**Writing – original draft:** Beminate Lemma Seifu, Getayeneh Antehunegn Tesema, Bezawit Melak Fentie, Tirualem Zeleke Yehuala, Abdulkerim Hassen Moloro, Kusse Urmale Mare.

**Writing – review & editing:** Beminate Lemma Seifu, Getayeneh Antehunegn Tesema, Bezawit Melak Fentie, Tirualem Zeleke Yehuala, Abdulkerim Hassen Moloro, Kusse Urmale Mare.

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
