## [Decision Letter · Decision Letter 0]

28 Sep 2023

PONE-D-23-24168Geographical variation in hotspots of stunting among under-five children in Ethiopia using the 2019 Demographic and Health Survey: a geographically weighted regression and multilevel robust Poisson regression analysisPLOS ONE

Dear Dr. Seifu,

Thank you for submitting your manuscript to PLOS ONE. After careful consideration, we feel that it has merit but does not fully meet PLOS ONE’s publication criteria as it currently stands. Therefore, we invite you to submit a revised version of the manuscript that addresses the points raised during the review process.

The manuscript has been evaluated by two reviewers, and their comments are available below. The reviewers have raised a number of concerns. They feel the manuscript would benefit from expanding the introduction to comprehensively summarize the current state of literature, updating the methods to provide a rational and more detailed description regarding the GWR analysis and the Poisson regression analysis, and expanding the discussion to contextualize the study results. Could you please carefully revise the manuscript to address all comments raised?  I also note that one or more reviewers has recommended that you cite specific previously published works. As always, we recommend that you please review and evaluate the requested works to determine whether they are relevant and should be cited. It is not a requirement to cite these works. We appreciate your attention to this request.

We look forward to receiving your revised manuscript.

Kind regards,

Johanna Pruller, PhD

Associate Editor

PLOS ONE

- https://doi.org/10.1371/journal.pone.0259147

- https://doi.org/10.1186/s12916-020-01786-5

- 10.3233/SJI-200717

In your revision ensure you cite all your sources (including your own works), and quote or rephrase any duplicated text outside the methods section. Further consideration is dependent on these concerns being addressed.

4. We note that Figures 2,4,5,6,7 and 8 in your submission contain [map/satellite] images which may be copyrighted. All PLOS content is published under the Creative Commons Attribution License (CC BY 4.0), which means that the manuscript, images, and Supporting Information files will be freely available online, and any third party is permitted to access, download, copy, distribute, and use these materials in any way, even commercially, with proper attribution. For these reasons, we cannot publish previously copyrighted maps or satellite images created using proprietary data, such as Google software (Google Maps, Street View, and Earth). For more information, see our copyright guidelines: http://journals.plos.org/plosone/s/licenses-and-copyright.

a. You may seek permission from the original copyright holder of Figures 2,4,5,6,7 and 8 to publish the content specifically under the CC BY 4.0 license. 

Reviewers' comments:

Reviewer's Responses to Questions

**Comments to the Author**

1. Is the manuscript technically sound, and do the data support the conclusions?

Reviewer #1: Partly

Reviewer #2: Yes

2. Has the statistical analysis been performed appropriately and rigorously? 

Reviewer #1: Yes

Reviewer #2: Yes

3. Have the authors made all data underlying the findings in their manuscript fully available?

Reviewer #1: Yes

Reviewer #2: Yes

4. Is the manuscript presented in an intelligible fashion and written in standard English?

Reviewer #1: Yes

Reviewer #2: Yes

5. Review Comments to the Author

Reviewer #1: In this study, our aim was to investigate spatial disparities in the prevalence of stunted growth among children under the age of five in Ethiopia. Additionally, we explored how socioeconomic factors vary across space and influence stunted growth in the same context. While the topic under consideration is of significant importance, our study lacks a comprehensive contextualization within the existing body of research, particularly with respect to similar studies conducted using comparable methods in the same country. It is imperative to elucidate the unique contributions our study brings to the field of research. Consequently, substantial enhancements are required in the major sections of the paper to align it with the standards necessary for publication. Please refer to the detailed comments provided below for specific improvements.

Introduction

1. Providing Context and Rationale for the Study: Considering the existing body of research in Ethiopia on stunted growth, which often employs hotspot analysis as a common methodological approach, it is crucial to establish the unique contributions of our study. Furthermore, the absence of citations to these pertinent studies in our paper raises questions about the comprehensiveness of our literature review, including those published within this journal. To rectify this, I recommend revisiting the introduction section to incorporate references to these studies and elucidate the rationale for our study. Please refer to the specific studies mentioned for guidance and justification.

Haile, D., Azage, M., Mola, T. et al. Exploring spatial variations and factors associated with childhood stunting in Ethiopia: spatial and multilevel analysis. BMC Pediatr 16, 49 (2016). https://doi.org/10.1186/s12887-016-0587-9

Kuse, K.A., Debeko, D.D. Spatial distribution and determinants of stunting, wasting and underweight in children under-five in Ethiopia. BMC Public Health 23, 641 (2023). https://doi.org/10.1186/s12889-023-15488-z

Hagos S, Hailemariam D, WoldeHanna T, Lindtjørn B (2017) Spatial heterogeneity and risk factors for stunting among children under age five in Ethiopia: A Bayesian geo-statistical model. PLoS ONE 12(2): e0170785. https://doi.org/10.1371/journal.pone.0170785

Muche A, Melaku MS, Amsalu ET, Adane M (2021) Using geographically weighted regression analysis to cluster under-nutrition and its predictors among under-five children in Ethiopia: Evidence from demographic and health survey. PLoS ONE 16(5): e0248156. https://doi.org/10.1371/journal.pone.0248156

2. Justifying the Importance of Spatial Heterogeneity: In the introduction, it is imperative to provide a compelling argument for the significance of investigating spatial heterogeneity in predictors of stunted growth. To bolster this justification, it is essential to draw upon insights from prior studies that have employed this method to explore stunted growth or malnutrition. Demonstrating how these findings have contributed to the field and explaining how they lend weight to our study's objectives will enhance the rationale for our research. For reference and context, please consider examining the studies mentioned.

Amegbor, P.M., Zhang, Z., Dalgaard, R. et al. Multilevel and spatial analyses of childhood malnutrition in Uganda: examining individual and contextual factors. Sci Rep 10, 20019 (2020). https://doi.org/10.1038/s41598-020-76856-y

Muche A, Melaku MS, Amsalu ET, Adane M (2021) Using geographically weighted regression analysis to cluster under-nutrition and its predictors among under-five children in Ethiopia: Evidence from demographic and health survey. PLoS ONE 16(5): e0248156. https://doi.org/10.1371/journal.pone.0248156

Biswas, M. Identifying Geographical Heterogeneity in Associations between Under-Five Child Nutritional Status and Its Correlates Across Indian Districts. Spat Demogr 10, 143–187 (2022). https://doi.org/10.1007/s40980-022-00104-2

3. Page 4 line 79 to 81: please provide a reference

Data and Methods

4. Measures for Spatial Analysis: It is essential for the authors to clearly define how stunted growth was measured at the Enumeration Area (EA) level in both the Ordinary Least Squares (OLS) and Geographically Weighted Regression (GWR) models. Specify whether stunted growth was expressed as a rate per EAs, an absolute count, or using another metric to eliminate ambiguity.

5. Modified Poisson Regression: The description of the use of modified Poisson regression in the manuscript should be revised for clarity. The current explanation is not easily comprehensible. I recommend consulting the provided reference for guidance on how to present the use of modified Poisson regression effectively.

Amegbor, P. M. (2022). Early-life environmental exposures and anaemia among children under age five in Sub-Saharan Africa: An insight from the Demographic & Health Surveys. Science of the Total Environment, 832, 154957.

6. Data Management and Analysis: The section related to data management and analysis should be revised to provide more detailed information, especially concerning the process of merging survey data with GPS data. It would be valuable to explain how this integration was conducted. You may refer to the mentioned references for insight on how to structure this section effectively.

7. Use of GWR: The authors should address the choice of Geographically Weighted Regression (GWR) over alternatives such as Multiscale Geographically Weighted Regression (MGWR) in light of the known limitations of GWR. Additionally, clarify whether fixed or adaptive bandwidth was employed and provide a rationale for this choice. Specify which search option was used for selecting the optimal bandwidth. To justify the choice, consider referring to the provided references for further insights.

Fotheringham, A. S., Yang, W. & Kang, W. Multiscale geographically weighted regression (MGWR). Ann. Am. Assoc. Geogr. 107, 1247–1265 (2017).

Amegbor, P.M., Zhang, Z., Dalgaard, R. et al. Multilevel and spatial analyses of childhood malnutrition in Uganda: examining individual and contextual factors. Sci Rep 10, 20019 (2020). https://doi.org/10.1038/s41598-020-76856-y

Also, provide the information on whether fixed or adaptive bandwidth was used and why? And which search option was used for selecting the optimal bandwidth.

Results

8. Page 13, Lines 205-207: It is recommended to rewrite the sentence for clarity. Specify which models are being compared, as the current wording is confusing. While the Mixed Effect Poisson model is mentioned, it's unclear which other model or models are being referred to.

9. Table 3: Regarding Table 3, the choice of categorizing mother age below 20 should be explained, especially considering there was a significant association for mother age above 40 in the Mixed Effect Poisson model. Additionally, clarify that the table represents the proportion of mothers in poor households, as suggested.

Discussion

10. Grounding the Discussion: In the discussion section, it's important to contextualize the study's findings within the broader Sub-Saharan Africa (SSA) region, rather than referring to China. Given the prevalence of studies on stunted growth in SSA, consider comparing your findings with regional prevalence and explaining potential factors accounting for similarities, differences, or disparities. You may refer to the provided references for guidance on contextualizing your findings in the SSA context.

Amegbor, P.M., Sabel, C.E., Mortensen, L.H. et al. Early-life air pollution and green space exposures as determinants of stunting among children under age five in Sub-Saharan Africa. J Expo Sci Environ Epidemiol (2023). https://doi.org/10.1038/s41370-023-00572-8

Quamme, S. H., & Iversen, P. O. (2022). Prevalence of child stunting in Sub-Saharan Africa and its risk factors. Clinical Nutrition Open Science, 42, 49-61.

Tusting, L. S., Bradley, J., Bhatt, S., Gibson, H. S., Weiss, D. J., Shenton, F. C., & Lindsay, S. W. (2020). Environmental temperature and growth faltering in African children: a cross-sectional study. The Lancet Planetary Health, 4(3), e116-e123.

11. Regional Disparities: Provide explanations for the high prevalence of stunted growth in specific regions, such as Tigray, central, northeast, and southern Amhara, north, west, and south Afar, and east SNNPR. Analyze local or regional factors that could account for these disparities to make the discussion more contextualized and specific.

12. Interpreting GWR Results: Avoid using the term "hotspots" when discussing GWR results, as the method does not conceptually measure hotspots. Instead, focus on explaining the spatial variation in associations between predictors and stunted growth in a more methodologically accurate manner.

13. GWR Results: Analyze the GWR results in-depth, particularly the positive associations of all predictors with stunted growth, except for poverty and male household heads. If maternal education and other factors had a global effect, consider exploring alternatives like Spatially Varying Coefficient Models (e.g., SGWR or Semi-Parametric GWR) in the absence of MGWR.

14. Contextualize GWR Discussion: Given that poverty and male household heads show both positive and negative associations in different areas, delve into why such spatial variations in association exist. Explore the reasons behind the positive association in some places and negative association in others, emphasizing the spatial variation in association, which is a central motivation for the study.

Reviewer #2: Dear Author,

Please consider the following observations and revise accordingly:

1. The author should correct this manuscript: In Ethiopia Stunting has steadily decreased, from 58% 71 in 2000 (2) to 36.81% in 2019 (3).

2. In the Introduction section: there is no explanation of the geographical conditions of stunting hotspots in Ethiopia.

3. What are the limitations of using DHS secondary data? And how to overcome these limitations?

4. How to calculate the sample size of 5490 children under five? What are the author's considerations in calculating the sample size?

5. The author corrected the punctuation in this manuscript: Dependent variable: In this study, the dependent variable was stunting, which was classified as "Yes = 1" for a child with a length or height/age -2 Z score and "No = 0" for a child with a height/age > -2 Z score.

6. Ethical considerations: How did the authors obtain ethical considerations for the use of secondary data? For example, for permissions and approval of data use.

7. On discussion: On discussion: what recommendations are given by the author in stunting prevention based on the findings of this study?

8. How does it compare with the results of previous studies?

6. PLOS authors have the option to publish the peer review history of their article (what does this mean?). If published, this will include your full peer review and any attached files.

Reviewer #1: **Yes: **Prince M. Amegbor

Reviewer #2: No

---

## [Author Response · Author response to Decision Letter 0]

27 Nov 2023

Point-by-point response 

Point by point response for editors/reviewers comments 

PLOS ONE

Manuscript title: Geographical variation in hotspots of stunting among under-five children in Ethiopia using the 2019 Demographic and Health Survey: a geographically weighted regression and multilevel robust Poisson regression analysis

Manuscript ID: PONE-D-23-24168

 Dear editor/reviewer. 

Dear all,

We would like to thank you for the constructive, building, and improvable comments on this manuscript that would improve the content of the manuscript. We considered each comment and clarification question of editors and reviewers on the manuscript thoroughly. Our point-by-point responses for each comment and question are described in detail on the following pages. Further, the details of changes were shown by track changes in the supplementary document attached

'Response to Reviewers

 Editor Comments:

https://journals.plos.org/plosone/s/file?id=wjVg/PLOSOne_formatting_sample_main_body.pdf andhttps://journals.plos.org/plosone/s/file?id=ba62/PLOSOne_formatting_sample_title_authors_affiliations.pdf

Authors’ response: Thank you dear editor for the comment. We have addressed. 

- https://doi.org/10.1371/journal.pone.0259147

- https://doi.org/10.1186/s12916-020-01786-5

- 10.3233/SJI-200717 

Authors’ response: Thank you dear editor for the comment. We have addressed. 

3. We note that you have stated that you will provide repository information for your data at acceptance. Should your manuscript be accepted for publication, we will hold it until you provide the relevant accession numbers or DOIs necessary to access your data. If you wish to make changes to your Data Availability statement, please describe these changes in your cover letter and we will update your Data Availability statement to reflect the information you provide

Authors’ response: Thank you dear editor for the comment. Data is available online and can be accessed from www.measuredhs.com.

4. We note that Figures 2,4,5,6,7 and 8 in your submission contain [map/satellite] images which may be copyrighted. All PLOS content is published under the Creative Commons Attribution License (CC BY 4.0), which means that the manuscript, images, and Supporting Information files will be freely available online, and any third party is permitted to access, download, copy, distribute, and use these materials in any way, even commercially, with proper attribution. For these reasons, we cannot publish previously copyrighted maps or satellite images created using proprietary data, such as Google software (Google Maps, Street View, and Earth). For more information, see our copyright guidelines: http://journals.plos.org/plosone/s/licenses-and-copyright.

Authors’ response: Thank you, editor, for seeking clarification. The maps reported in our study are not copyrighted from somewhere rather they are the output of work that we did using Arc-GIS version 10.8 statistical software based on the Ethiopian shapefile obtained from the Ethiopian Statistical Agency (CSA), which is publicly available on Open-Africa https://www.diva-gis.org/gdata and the GPS data obtained from the measure DHS program after requesting by uploading the rationales of the study. Therefore, we kindly assure you that the maps reported in the paper are our work produced through applying appropriate analytical methods and procedures. Besides, we have acknowledged the source of the data and we authors have several published studies that contained maps like these.

Response to Reviewer-1

Introduction

1. Providing Context and Rationale for the Study: Considering the existing body of research in Ethiopia on stunted growth, which often employs hotspot analysis as a common methodological approach, it is crucial to establish the unique contributions of our study. Furthermore, the absence of citations to these pertinent studies in our paper raises questions about the comprehensiveness of our literature review, including those published within this journal. To rectify this, I recommend revisiting the introduction section to incorporate references to these studies and elucidate the rationale for our study. Please refer to the specific studies mentioned for guidance and justification.

Haile, D., Azage, M., Mola, T. et al. Exploring spatial variations and factors associated with childhood stunting in Ethiopia: spatial and multilevel analysis. BMC Pediatr 16, 49 (2016). https://doi.org/10.1186/s12887-016-0587-9

Kuse, K.A., Debeko, D.D. Spatial distribution and determinants of stunting, wasting and underweight in children under-five in Ethiopia. BMC Public Health 23, 641 (2023). https://doi.org/10.1186/s12889-023-15488-z

Hagos S, Hailemariam D, WoldeHanna T, Lindtjørn B (2017) Spatial heterogeneity and risk factors for stunting among children under age five in Ethiopia: A Bayesian geo-statistical model. PLoS ONE 12(2): e0170785. https://doi.org/10.1371/journal.pone.0170785

Muche A, Melaku MS, Amsalu ET, Adane M (2021) Using geographically weighted regression analysis to cluster under-nutrition and its predictors among under-five children in Ethiopia: Evidence from demographic and health survey. PLoS ONE 16(5): e0248156. https://doi.org/10.1371/journal.pone.0248156

Authors’ response: Thank you dear reviewer for the constructive comments. We have done literature reviews and have cited all previously done studies regarding spatial distribution of stunting. Furthermore, we have include the uniqueness and the contribution of our study to the existing body of knowledge (please see the revised manuscript). 

2. Justifying the Importance of Spatial Heterogeneity: In the introduction, it is imperative to provide a compelling argument for the significance of investigating spatial heterogeneity in predictors of stunted growth. To bolster this justification, it is essential to draw upon insights from prior studies that have employed this method to explore stunted growth or malnutrition. Demonstrating how these findings have contributed to the field and explaining how they lend weight to our study's objectives will enhance the rationale for our research. For reference and context, please consider examining the studies mentioned.

Amegbor, P.M., Zhang, Z., Dalgaard, R. et al. Multilevel and spatial analyses of childhood malnutrition in Uganda: examining individual and contextual factors. Sci Rep 10, 20019 (2020). https://doi.org/10.1038/s41598-020-76856-y

Muche A, Melaku MS, Amsalu ET, Adane M (2021) Using geographically weighted regression analysis to cluster under-nutrition and its predictors among under-five children in Ethiopia: Evidence from demographic and health survey. PLoS ONE 16(5): e0248156. https://doi.org/10.1371/journal.pone.0248156

Biswas, M. Identifying Geographical Heterogeneity in Associations between Under-Five Child Nutritional Status and Its Correlates Across Indian Districts. Spat Demogr 10, 143–187 (2022). https://doi.org/10.1007/s40980-022-00104-2

Authors’ response: Thank you dear reviewer for the comment. We have used the references you suggest and include the rationale of studying spatial heterogeneity regarding the predictors of stunting among under five children. 

3. Page 4 line 79 to 81: please provide a reference 

Authors’ response: Thank you dear reviewer for the comment. We have cited the references. 

Data and Methods

4. Measures for Spatial Analysis: It is essential for the authors to clearly define how stunted growth was measured at the Enumeration Area (EA) level in both the Ordinary Least Squares (OLS) and Geographically Weighted Regression (GWR) models. Specify whether stunted growth was expressed as a rate per EAs, an absolute count, or using another metric to eliminate ambiguity.

Authors’ response: Thank you dear reviewer for the comment. The proportion of stunted children among those under the age of five at the Enumeration Area (EA) level used as the dependent variable for spatial regression analysis. We have included this statement in the revised manuscript (Please see the revised manuscript) 

5. Modified Poisson Regression: The description of the use of modified Poisson regression in the manuscript should be revised for clarity. The current explanation is not easily comprehensible. I recommend consulting the provided reference for guidance on how to present the use of modified Poisson regression effectively.

Amegbor, P. M. (2022). Early-life environmental exposures and anaemia among children under age five in Sub-Saharan Africa: An insight from the Demographic & Health Surveys. Science of the Total Environment, 832, 154957.

Authors’ response: Thank you dear reviewer for the comment. We have addressed the raised comment based on the suggested reference (see the revised manuscript). 

6. Data Management and Analysis: The section related to data management and analysis should be revised to provide more detailed information, especially concerning the process of merging survey data with GPS data. It would be valuable to explain how this integration was conducted. You may refer to the mentioned references for insight on how to structure this section effectively.

Authors’ response: Thank you dear reviewer for the comment. The outcome and covariates were collected at individual level whereas the geographic coordinate data (latitude and longitude) at cluster level/Enumeration Areas (EAs) level. For the GWR, we have aggregated the outcome and covariates at EA level and then merged with the GPS data using cluster number/EAs as merging variable. This was the data management procedure to prepare the data for the spatial analysis.

7. Use of GWR: The authors should address the choice of Geographically Weighted Regression (GWR) over alternatives such as Multiscale Geographically Weighted Regression (MGWR) in light of the known limitations of GWR. Additionally, clarify whether fixed or adaptive bandwidth was employed and provide a rationale for this choice. Specify which search option was used for selecting the optimal bandwidth. To justify the choice, consider referring to the provided references for further insights.

Fotheringham, A. S., Yang, W. & Kang, W. Multiscale geographically weighted regression (MGWR). Ann. Am. Assoc. Geogr. 107, 1247–1265 (2017).

Amegbor, P.M., Zhang, Z., Dalgaard, R. et al. Multilevel and spatial analyses of childhood malnutrition in Uganda: examining individual and contextual factors. Sci Rep 10, 20019 (2020). https://doi.org/10.1038/s41598-020-76856-y

Also, provide the information on whether fixed or adaptive bandwidth was used and why? And which search option was used for selecting the optimal bandwidth.

Authors’ response: Thank you reviewer for the comment. Initially we have fitted the Ordinary Least Square regression (OLS) and the Koenker (BP) Statistic test was statistically significant (p < 0.01), that means the relationships modeled are not consistent (either due to non-stationarity or heteroskedasticity). And then we have fitted GWR and MGWR models. Then compared using corrected AIC (AICc) and Adjusted R2. Based on this GWR had lower AICc and higher Adjusted R2 values compared to MGWR, that is why we have reported the GWR results. As you know GWR assumes the coefficients to vary over space at the same spatial scale. Whereas, the MGWR the coefficients to vary over space but also allows the scale to vary across different explanatory variables. For optimal bandwidth, we used adaptive bandwidth as it outweighs the fixed bandwidth. 

8. Page 13, Lines 205-207: It is recommended to rewrite the sentence for clarity. Specify which models are being compared, as the current wording is confusing. While the Mixed Effect Poisson model is mentioned, it's unclear which other model or models are being referred to.

Authors’ response: Thank you dear reviewer for the comment. We have fitted four models separately. Model 1 (null model) was fitted without independent variables to estimate the cluster-level variation of stunting in Ethiopia. Model 2 and Model 3 were adjusted for individual-level variables and community-level variables, respectively. Model 4 was the final model adjusted for individual and community-level variables simultaneously (please see the method part of the revised manuscript).

9. Table 3: Regarding Table 3, the choice of categorizing mother age below 20 should be explained, especially considering there was a significant association for mother age above 40 in the Mixed Effect Poisson model. Additionally, clarify that the table represents the proportion of mothers in poor households, as suggested.

Authors’ response: Thank you dear reviewer for the comment. Given that maternal age above 40 has been identified as a protective factor compared to maternal age below 20, which is a known risk factor, it is imperative that we analyze the geographical distribution of a variable that is indicative of risk. The individual-level data on wealth indexes are aggregated to the EA level. This involves summing up the counts of “poorer” and “poor” households within each EA. The proportion of poor households is obtained by dividing the aggregated count of “poorer” and “poor” households by the total population in that EA.

Discussion

10. Grounding the Discussion: In the discussion section, it's important to contextualize the study's findings within the broader Sub-Saharan Africa (SSA) region, rather than referring to China. Given the prevalence of studies on stunted growth in SSA, consider comparing your findings with regional prevalence and explaining potential factors accounting for similarities, differences, or disparities. You may refer to the provided references for guidance on contextualizing your findings in the SSA context.

Amegbor, P.M., Sabel, C.E., Mortensen, L.H. et al. Early-life air pollution and green space exposures as determinants of stunting among children under age five in Sub-Saharan Africa. J Expo Sci Environ Epidemiol (2023). https://doi.org/10.1038/s41370-023-00572-8

Quamme, S. H., & Iversen, P. O. (2022). Prevalence of child stunting in Sub-Saharan Africa and its risk factors. Clinical Nutrition Open Science, 42, 49-61.

Tusting, L. S., Bradley, J., Bhatt, S., Gibson, H. S., Weiss, D. J., Shenton, F. C., & Lindsay, S. W. (2020). Environmental temperature and growth faltering in African children: a cross-sectional study. The Lancet Planetary Health, 4(3), e116-e123.

Authors’ response: Thank you dear reviewer for the comment. We have tried to revise the discussion part regarding the prevalence of stunting in Ethiopia compared to other SSA countries, but as per our literature review, the prevalence of stunting across African countries is relatively consistent. We have cited those studies that have been done in SSA among children under the age of five. 

11. Regional Disparities: Provide explanations for the high prevalence of stunted growth in specific regions, such as Tigray, central, northeast, and southern Amhara, north, west, and south Afar, and east SNNPR. Analyse local or regional factors that could account for these disparities to make the discussion more contextualized and specific.

Author’s response: Dear reviewer thank you for your comment. We have exhaustively discussed the local and regional factors that could account for high proportion of stunting in those regions. (Please see the revised manuscript)

12. Interpreting GWR Results: Avoid using the term "hotspots" when discussing GWR results, as the method does not conceptually measure hotspots. Instead, focus on explaining the spatial variation in associations between predictors and stunted growth in a more methodologically accurate manner.

Author’s response: Thank you for the comment. We have addressed it.

13. GWR Results: Analyze the GWR results in-depth, particularly the positive associations of all predictors with stunted growth, except for

---

## [Decision Letter · Decision Letter 1]

18 Feb 2024

PONE-D-23-24168R1Geographical variation in hotspots of stunting among under-five children in Ethiopia using the 2019 Demographic and Health Survey: a geographically weighted regression and multilevel robust Poisson regression analysisPLOS ONE

Dear Dr. Seifu,

Thank you for submitting your manuscript to PLOS ONE. After careful consideration, we feel that it has merit but does not fully meet PLOS ONE’s publication criteria as it currently stands. Therefore, we invite you to submit a revised version of the manuscript that addresses the points raised during the review process.

We look forward to receiving your revised manuscript.

Kind regards,

Johanna Pruller, Ph.D.

Associate Editor

PLOS ONE

**Additional Editor Comments:**

The manuscript has been evaluated by two reviewers, and one reviewer has major concerns.

Could you please carefully revise the manuscript to address all comments raised. In particular, could you please pay special attention to the second comment raised by the reviewer, and expand on how you conducted the MGWR test?

Reviewers' comments:

Reviewer's Responses to Questions

**Comments to the Author**

1. If the authors have adequately addressed your comments raised in a previous round of review and you feel that this manuscript is now acceptable for publication, you may indicate that here to bypass the “Comments to the Author” section, enter your conflict of interest statement in the “Confidential to Editor” section, and submit your "Accept" recommendation.

Reviewer #1: (No Response)

Reviewer #2: All comments have been addressed

2. Is the manuscript technically sound, and do the data support the conclusions?

Reviewer #1: Yes

Reviewer #2: Yes

3. Has the statistical analysis been performed appropriately and rigorously? 

Reviewer #1: No

Reviewer #2: Yes

4. Have the authors made all data underlying the findings in their manuscript fully available?

Reviewer #1: Yes

Reviewer #2: Yes

5. Is the manuscript presented in an intelligible fashion and written in standard English?

Reviewer #1: Yes

Reviewer #2: Yes

6. Review Comments to the Author

Reviewer #1: While some of the initial concerns have been addressed, others remain unattended. Please find my detailed comments below.

Methods:

1. Sample Size: In accordance with Reviewer 2’s comments, the authors should provide information on how they arrived at the final sample size of 5,490 (Page 6, lines 126 to 127).

2. Comment 7 – Use of GWR: In response to my initial comment on the limitations of GWR and the advice for the authors to consider MGWR, the authors claim they used AUCc and R2 to determine the best-fitting model. To the best of my knowledge, the analytic platform used for spatial analysis, ARCGIS 10.7, doesn’t support MGWR testing or lacks MGWR functionality. Thus, how did the authors conduct the MGWR test? Additionally, the authors should provide the test diagnostics information for both the GWR and MGWR models in their response. Note that the developer of GWR strongly recommends the use of MGWR due to the known limitations of GWR; hence, I encourage the authors to read the Fotheringham et al. paper referenced in comment 7 in my previous assessment.

Discussion:

3. Page 21, lines 323 to 324: Please clarify this sentence: “The prevalence of stunting 324 is consistent with a 33 (8) and 55 (30) SSA countries and East African study (31).” It is incomprehensible in its current form.

4. Comment 12 – Use of Hotspot for GWR Result Interpretation: Despite my previous caution against using the term "hotspot" for GWR results, it still appears in the revised manuscript (e.g., Page 23, lines 366). Please refrain from using "hotspot" in reference to GWR results. Refer to the suggested reference for the correct interpretation of the GWR results.

5. Discussion of Poverty and Stunted Growth GWR Results: The authors need to clarify whether the intervention programs cited as offsetting the effect of poverty in some areas are specifically for those communities and provide a reference to that effect. If they are nationwide programs, then the authors should explain why they affect some communities positively and have no effect in other communities.

Reviewer #2: Dear author, thank you for revising the manuscript based on the reviewer's comments. The revisions you made in response to the feedback provided further strengthened the quality of the manuscript.

7. PLOS authors have the option to publish the peer review history of their article (what does this mean?). If published, this will include your full peer review and any attached files.

Reviewer #1: **Yes: **Prince M. Amegbor

Reviewer #2: No

---

## [Author Response · Author response to Decision Letter 1]

26 Feb 2024

Point-by-point response 

Point by point response for editors/reviewers comments 

PLOS ONE

Manuscript title: Geographical variation in hotspots of stunting among under-five children in Ethiopia using the 2019 Demographic and Health Survey: a geographically weighted regression and multilevel robust Poisson regression analysis

Manuscript ID: PONE-D-23-24168

 Dear editor/reviewer. 

Dear all,

We would like to thank you for the constructive, building, and improvable comments on this manuscript that would improve the content of the manuscript. We considered each comment and clarification question of editors and reviewers on the manuscript thoroughly. Our point-by-point responses for each comment and question are described in detail on the following pages. Further, the details of changes were shown by track changes in the supplementary document attached

Additional Editor Comments:

The manuscript has been evaluated by two reviewers, and one reviewer has major concerns.

Could you please carefully revise the manuscript to address all comments raised. In particular, could you please pay special attention to the second comment raised by the reviewer, and expand on how you conducted the MGWR test?

Author’s response: Dear Academic editor thank you for your concerns. As mentioned by the reviewer, ArcGIS does not support MGWR. Therefore, we used the MGWR 2.2 statistical software to run the MGWR test for our dataset. Since GWR was the best-fitted model for our data, we ran the GWR model using ArcGIS software. We have included the results of the model diagnostic tests below.

'Response to Reviewers

Methods:

1. Sample Size: In accordance with Reviewer 2’s comments, the authors should provide information on how they arrived at the final sample size of 5,490 (Page 6, lines 126 to 127). 

Author’s response: Given that we already utilized a nationally representative dataset with an appropriate sample size, recalculating the sample size seems unnecessary. As you know, DHS surveys are representative at the national level, for urban and rural areas. All survey-sampling strategies are subject to sampling error. The DHS Program designs samples to provide national and subnational estimates with a reasonable relative standard error. The larger the sample size, the smaller the relative standard error on any given indicator will be. Therefore, we do not calculate sample size while we use this data we used the available sample as it is. By utilizing the available sample as-is, we are leveraging the existing data efficiently. This approach ensures that the estimates are robust while minimizing unnecessary recalculations.

2. Comment 7 – Use of GWR: In response to my initial comment on the limitations of GWR and the advice for the authors to consider MGWR, the authors claim they used AUCc and R2 to determine the best-fitting model. To the best of my knowledge, the analytic platform used for spatial analysis, ARCGIS 10.7, doesn’t support MGWR testing or lacks MGWR functionality. Thus, how did the authors conduct the MGWR test? Additionally, the authors should provide the test diagnostics information for both the GWR and MGWR models in their response. Note that the developer of GWR strongly recommends the use of MGWR due to the known limitations of GWR; hence, I encourage the authors to read the Fotheringham et al. paper referenced in comment 7 in my previous assessment 

Author’s response: Dear reviewer thank you for your suggestion. Given that ArcGIS does not support MGWR, we employed the MGWR 2.2 statistical software to conduct the MGWR test. Notably, we found the GWR model to be the best-fitted model for our dataset. Consequently, we utilized ArcGIS software to run the GWR model.

Here is the test diagnostic information for both models from the MGWR 2.2 software

diagnostic information GWR MGWR

AIC 457.894 778.972

AICc 459.482 781.815

BIC 512.459 851.930

Adjusted R-square 0.37 0.29

Discussion:

3. Page 21, lines 323 to 324: Please clarify this sentence: “The prevalence of stunting 324 is consistent with a 33 (8) and 55 (30) SSA countries and East African study (31).” It is incomprehensible in its current form. 

Author’s response: Dear reviewer thank you for the comment. We have rewrite the aforementioned sentence. (Please see the revised manuscript)

4. Comment 12 – Use of Hotspot for GWR Result Interpretation: Despite my previous caution against using the term "hotspot" for GWR results, it still appears in the revised manuscript (e.g., Page 23, lines 366). Please refrain from using "hotspot" in reference to GWR results. Refer to the suggested reference for the correct interpretation of the GWR results. 

Author’s response: Dear reviewer thank you for the comment. It was an editorial mistake we have addressed it. 

5. Discussion of Poverty and Stunted Growth GWR Results: The authors need to clarify whether the intervention programs cited as offsetting the effect of poverty in some areas are specifically for those communities and provide a reference to that effect. If they are nationwide programs, then the authors should explain why they affect some communities positively and have no effect in other communities.

Author’s response: Dear reviewer thank you for the comment. We have provided a reference for nutritional interventions and policy implementations that apply to specific regions. Additionally, we have included possible reasons for why these interventions have diverse outcomes in different regions. For future researchers, it would be extremely valuable to investigate the reasons behind the varying effects of these interventions in different regions.

---

## [Decision Letter · Decision Letter 2]

24 Mar 2024

PONE-D-23-24168R2Geographical variation in hotspots of stunting among under-five children in Ethiopia using the 2019 Demographic and Health Survey: a geographically weighted regression and multilevel robust Poisson regression analysisPLOS ONE

Dear Dr. Seifu,

Thank you for submitting your manuscript to PLOS ONE. After careful consideration, we feel that it has merit but does not fully meet PLOS ONE’s publication criteria as it currently stands. Therefore, we invite you to submit a revised version of the manuscript that addresses the points raised during the review process.

We look forward to receiving your revised manuscript.

Kind regards,

Dereje Oljira Donacho, PhD in Environmental Health

Academic Editor

PLOS ONE

Journal Requirements:

Reviewers' comments:

Reviewer's Responses to Questions

**Comments to the Author**

1. If the authors have adequately addressed your comments raised in a previous round of review and you feel that this manuscript is now acceptable for publication, you may indicate that here to bypass the “Comments to the Author” section, enter your conflict of interest statement in the “Confidential to Editor” section, and submit your "Accept" recommendation.

Reviewer #1: All comments have been addressed

Reviewer #3: (No Response)

2. Is the manuscript technically sound, and do the data support the conclusions?

Reviewer #1: Yes

Reviewer #3: Yes

3. Has the statistical analysis been performed appropriately and rigorously? 

Reviewer #1: Yes

Reviewer #3: Yes

4. Have the authors made all data underlying the findings in their manuscript fully available?

Reviewer #1: Yes

Reviewer #3: Yes

5. Is the manuscript presented in an intelligible fashion and written in standard English?

Reviewer #1: Yes

Reviewer #3: No

6. Review Comments to the Author

Reviewer #1: (No Response)

Reviewer #3: Comments to Authors

Thank you for addressing such interesting public health concern.

Abstract

Line 46…. rich household wealth index. I suggest to replace this phrase with “household wealth status”.

Method and materials

Line 141… community maternal illiteracy level… please use maternal literacy level instead of maternal illiteracy level.

Line 179: Did the authors check the assumption of spatial regression analysis?

Results

Table 1. The way by which authors compute household wealth index is not explained in methodology part. I suggest the authors explain how many variables were used to compute household wealth index? Did you check the assumption of PCA?...... need to be explained in the methodology section.

Table: 1 Source of drinking water

Improved, not improved… What is improved and not source of drinking water? I suggest these terms need to be operationalized.

Table :3

Proportion of mothers with poverty…. please replace this phrase with “proportion of mother with poor household wealth status”

Discussion

The discussion is shallow, and need to be discussed by adding more previous studies across the world.

Line 338…Tigray has been affected by armed conflict, leading to displacement, disrupted healthcare services, and food insecurity... As to me this cannot be a justification because the war in Tigray region was erupted in November 2020. The authors used 2019 EMDHS data which is before the war. Thus, the consequence of Tigray armed conflict cannot justify the discrepancy.

-Significantly associated variables in the geographically weighted regression were not well discussed. I suggest the authors compare this finding by adding more previous studies across the countries.

Conclusion

-Please clearly add the implication of the finding.

7. PLOS authors have the option to publish the peer review history of their article (what does this mean?). If published, this will include your full peer review and any attached files.

Reviewer #1: **Yes: **Prince Amegbor

Reviewer #3: No

---

## [Author Response · Author response to Decision Letter 2]

4 Apr 2024

Point-by-point response 

Point by point response for editors/reviewers comments 

PLOS ONE

Manuscript title: Geographical variation in hotspots of stunting among under-five children in Ethiopia using the 2019 Demographic and Health Survey: a geographically weighted regression and multilevel robust Poisson regression analysis

Manuscript ID: PONE-D-23-24168R2

 Dear editor/reviewer. 

Dear all,

We would like to thank you for the constructive, building, and improvable comments on this manuscript that would improve the content of the manuscript. We considered each comment and clarification question of editors and reviewers on the manuscript thoroughly. Our point-by-point responses for each comment and question are described in detail on the following pages. Further, the details of changes were shown by track changes in the supplementary document attached

'Response to Reviewers

Abstract 

Line 46…. rich household wealth index. I suggest to replace this phrase with “household wealth status”.

Author’s response: Dear reviewer thank you for your comment. We have replaced the phrase with “household wealth status”.

Method and materials

Line 141… community maternal illiteracy level… please use maternal literacy level instead of maternal illiteracy level.

Author’s response: Dear reviewer thank you for your comment. We have replaced the phrase with “maternal literacy level”.

Line 179: Did the authors check the assumption of spatial regression analysis? 

Author’s response: Before proceeding to the local model, the six assumptions of the OLS model (the explanatory variables should have the expected relationship, the significance of each explanatory variable, the randomness of residuals, assuring the statistical no significance of Jarque-Bera statistics, Variance Inflation Factor (VIF) value, and the strength of R-square) were checked. The results for the tests of those assumptions were presented in the manuscript (Table 3 and 4). 

 Results 

Table 1. The way by which authors compute household wealth index is not explained in methodology part. I suggest the authors explain how many variables were used to compute household wealth index? Did you check the assumption of PCA?...... need to be explained in the methodology section.

Author’s response: we do not compute household wealth index because in the Demographic and Health Surveys (DHS), the household wealth index is indeed constructed using household asset data via principal components analysis (PCA). This index provides a measure of relative wealth or socioeconomic status for each household surveyed. 

https://www.dhsprogram.com/topics/wealth-index/Wealth-Index-Construction.cfm

Table: 1 Source of drinking water 

 Improved, not improved… What is improved and not source of drinking water? I suggest these terms need to be operationalized.

Author’s response: Dear reviewer thank you for the suggestion. We have included the operational definition for source of drinking water. 

Table :3

Proportion of mothers with poverty…. please replace this phrase with “proportion of mother with poor household wealth status”

Author’s response: Thank you dear reviewer for your comment. We have replaced the term with “proportion of mother with poor household wealth status”

Discussion 

The discussion is shallow, and need to be discussed by adding more previous studies across the world. 

Line 338…Tigray has been affected by armed conflict, leading to displacement, disrupted healthcare services, and food insecurity... As to me this cannot be a justification because the war in Tigray region was erupted in November 2020. The authors used 2019 EMDHS data which is before the war. Thus, the consequence of Tigray armed conflict cannot justify the discrepancy. 

-Significantly associated variables in the geographically weighted regression were not well discussed. I suggest the authors compare this finding by adding more previous studies across the countries.

Author’s response: Thank you dear reviewer for your comment. We have thoroughly revised our discussion by adding a multitude of pertinent previous studies. 

Conclusion 

-Please clearly add the implication of the finding.

Author’s response: Thank you dear reviewer for your comment. We have added implications

---

## [Editor Report · Decision Letter 3]

19 Apr 2024

Geographical variation in hotspots of stunting among under-five children in Ethiopia: a geographically weighted regression and multilevel robust Poisson regression analysis

PONE-D-23-24168R3

Dear Dr. Seifu,

We’re pleased to inform you that your manuscript has been judged scientifically suitable for publication and will be formally accepted for publication once it meets all outstanding technical requirements.

Kind regards,

Dereje Oljira Donacho, PhD in Environmental Health

Academic Editor

PLOS ONE
---

## [Editor Report · Acceptance letter]

26 Apr 2024

PONE-D-23-24168R3 

PLOS ONE

Dear Dr. Seifu, 

I'm pleased to inform you that your manuscript has been deemed suitable for publication in PLOS ONE. Congratulations! Your manuscript is now being handed over to our production team.

Kind regards, 

on behalf of

Dr. Dereje Oljira Donacho 

Academic Editor

PLOS ONE